# Methods to establish a Pregnancy Register in the QResearch Database
Andrew JHL Snelling [1], Emma Copland [1,2], Winnie X. Mei[2], Wema M. Mtika [2], Tom Ranger[2], Carol Coupland[1,3], QResearch Pregnancy Consortium*, Aziz Sheikh[2], Julia Hippisley-Cox [1] & Jennifer A. Hirst [2] ✉

## Abstract

**Background** Electronic health records are increasingly used to conduct pregnancy-related research as pregnant women are under-represented in research. Creating a register of pregnancies by combining data from primary and secondary care will further facilitate research in pregnancy. This work describes the construction of an algorithm to create a unified pregnancy cohort in the QResearch database during the emergency phase of the COVID-19 pandemic.

**Methods** National primary care records in the QResearch® database were linked to patient-level data from Hospital Episode Statistics (HES) datasets. Females aged 15-50 years with a pregnancy outcome recorded between 30 December 2020 and 30 September 2022 were included. Pregnancy (delivery/loss) episodes were identified and cohort demographics reported using a three-stage algorithm. Pregnancy start dates were derived using a combination of HES and primary care data, or individually estimated where no corresponding date could be identified.

**Results** 266,758 women with 279,027 pregnancies are captured in the register. 232,673 pregnancies (83.4%) are deliveries (99.5% livebirths and 0.5% stillbirths) and 46,354 (16.6%) pregnancies are pregnancy losses. Pregnancy losses are highest amongst those of Caribbean (23.1%; n = 781) ethnicity and lowest in those of Pakistani ethnicity (13.9%, n = 1,579). 82.4% of pregnancies are derived from HES maternity records, 10.6% from primary care records, 3.4% from HES Admissions, and 3.6% from HES Procedures.

**Conclusion** The construction of a pregnancy register in QResearch® offers a valuable resource for future research. Its methodology can be adapted to construct new cohorts over any period, providing a comprehensive resource on pregnancy outcomes and events.

## Plain language summary

Digital patient records are being used to study pregnant women more and more frequently. However, a method to create a single, unified list of pregnancies using both hospital and GP data has not yet been designed for the university-based research database, QResearch, which contains health records for over 40 million people in the UK. The aim of this study was to develop a method of combining and sorting pregnancy records from multiple data sources to create a list, or register, of pregnancies. The method successfully identified women who gave birth, terminated, or lost a pregnancy during the COVID-19 pandemic in England. Furthermore, this method can be adapted to identify pregnancies over any period, making it a valuable resource for researchers who are interested in studying pregnant women in the future.

Routinely collected electronic health records (EHRs) are used increasingly in medical research to determine healthcare usage, effect of treatments, and health outcomes in the general population and have the potential to follow-up populations over long periods of time[1]. EHRs also make it possible to study subsets of populations who are typically excluded from clinical trials and for whom the effects of treatments may not be well understood, for example, pregnant women[1]. Pregnant women are a group who are often excluded from clinical trials and for whom it would be particularly valuable to understand how co-morbidities and exposures (e.g. prescribed medication) may impact their health, pregnancy, and the health of their infants[2].

Being able to determine the scale of medication use or evaluate the impact of an exposure on pregnancy outcomes is important. For example, vaccination and medication use in pregnancy is common, but current knowledge of the clinical risk/benefit and safety profiles of many types of medications during pregnancy remains limited[3]. As a result, decisions around starting, continuing, and stopping treatments with medications during pregnancy are challenging for pregnant women and clinicians, with these decisions having the potential to impact the long-term health and well-being of women and their children. During the COVID-19 pandemic, a lack of data about COVID-19 vaccine safety in pregnancy and the possibility of harm to the

[1]Wolfson Institute of Population Health, Barts and The London School of Medicine and Dentistry, Queen Mary University of London, London, UK. [2]Nuffield Department of Primary Care Health Sciences, University of Oxford, Oxford, UK. [3]School of Medicine, University of Nottingham, Nottingham, UK. *A list of authors and their affiliations appears at the end of the paper. ✉e-mail: jennifer.hirst@phc.ox.ac.uk

fetus were the primary reasons for refusing vaccination in pregnant women[4]. This may have contributed to vaccine hesitancy found and reported by pregnant women and clinicians during the UK Government's COVID-19 Vaccination Programme[5]. This hesitancy may have resulted in preventable deaths from COVID-19 amongst unvaccinated pregnant women and their infants[6].

While there is evidence to suggest that there is no association between COVID-19 vaccination and an increased risk of perinatal death amongst pregnant women during the pandemic[7], there is a pressing need for datasets and tools to conduct further, robust research studies into the benefits and possible harms of exposures or interventions in pregnancy, their associated impact on the incidence of pregnancy outcomes (i.e. livebirth, stillbirth, miscarriage), and the relative importance of sociodemographic factors (i.e. ethnicity, social deprivation) in determining risk in the general population of pregnant women. EHRs have the potential to be such a dataset, as demonstrated by the Clinical Research Datalink (CPRD)[8–12] and Nordic registries[13–16]. However, there are known challenges in defining a pregnancy cohort using routine health records, in particular, determining the start date and duration of a pregnancy[9]. Pregnancy registers have been established in a number of existing UK EHR databases[8–12,17], however, at present no equivalent pregnancy register which can be used for research purposes has been developed in the QResearch Database[18]. QResearch is a UK primary care database covering approximately 20% of English primary care practices spread across all regions in England. At the time this research was conducted, it housed records from a total of 13 million patients registered with participating practices using the Optum electronic patient record system, EMIS Web, with a further 25 million historical patients with longitudinal linked data stretching back to the 1990s. It includes sociodemographic data with a high level of completeness[19]. Here, we report on the development of an algorithm to establish a pregnancy register in the QResearch database. We describe the methods used to create the register and the characteristics of a cohort of women who had a pregnancy during the emergency phase of COVID-19 pandemic, between 2020 and 2022. This work was undertaken as part of a project funded by the National Institute for Health and care Research (NIHR) School for Primary Care Research to evaluate uptake, safety, and effectiveness of COVID-19 vaccination during pregnancy[20].

The cohort contains 266,758 women with 279,027 pregnancies. 232,673 pregnancies (83.4%) are deliveries (99.5% live births and 0.5% stillbirths) and 46,354 (16.6%) pregnancies are pregnancy losses. The proportion of pregnancy losses and deliveries were comparable across all covariates, but differed most by ethnicity, region, and age group. Most pregnancies (82.4%) are derived from HES maternity records, 10.6% from primary care records, 3.4% from HES Admissions, and 3.6% from HES Procedures. The results of the supplementary analyses from the restricted cohort are comparable to the main analyses, with a 3–5% increase in the proportion of miscarriages and a commensurate drop in the proportion of terminations. The internal validation shows a high level of agreement between data sources for deliveries and a lower level of agreement for pregnancy losses. The results of the external validation shows that internal delivery/pregnancy loss estimates are, generally, in line with national and published estimates.

## Methods
### Study period
The study period was from 30 December 2020 (when pregnant women became eligible for COVID-19 vaccination[21]) to 30 September 2022 (date of last available data extract at the time of analysis).

### Study population
We identified individuals recorded as female in general practice (GP) records who were aged 15–50 years and registered with participating primary care practices in QResearch. Only those with a pregnancy outcome (delivery or pregnancy loss) recorded in Hospital Episode Statistics (HES) or GP data during the study period were included in the cohort.

### Data sources
This project used the QResearch primary care database with patient demographics, diagnoses, medication prescriptions, laboratory investigations, and referrals. These data were linked at the individual level to HES, consisting of Admitted Patient Care (APC) data (admission level data for National Health Service (NHS) hospital admissions in the UK), HES Maternity fields (a subset of HES APC used to characterise and define hospital pregnancy episodes), and HES Office of Population Censuses and Surveys (OPCS) Classification of Interventions and Procedures, which contains surgical procedure codes which were used to identify procedures related to pregnancy[22]. The shorthand, HES Procedures and HES Admissions, are also used to refer to HES OPCS and HES APC, respectively, in this paper. Data in all HES datasets covered the entire span of the study period until 30 Sept 2022, while primary care data were available to 11 February 2022.

### Generating primary care (GP) and secondary care (HES) code lists
Electronic clinical code lists from primary care records (SNOMED CT), hospital diagnoses (International Statistical Classification of Diseases and Related Health Problems, 10th Revision (ICD-10)), and procedures were generated for all pregnancy-related visits using the QResearch data access platform, QWeb[23]. This was achieved by using codes included in previous studies[24], by mapping to code lists used to generate the CPRD pregnancy register[8–10], and in consultation with general practitioners, obstetricians, and midwives (Fig. 1—Stage 1). All primary and secondary codegroup lists may be found in Supplementary Data 1 (GP), 2 (ICD-10), and 3 (OPCS). The code lists were used to identify a population who had a pregnancy outcome recorded during the study period from primary and secondary care records. The primary care records were linked with HES and OPCS data, which enabled an algorithm to be developed to determine the pregnancy start dates, end dates, and define the outcome of each pregnancy. A unique personal identifier (patid) derived from a patient's NHS number using a one-way hashing algorithm was used to link patient data across all datasets used to construct the cohort. The eligible population ($n = 595,787$) was defined as those having a known pregnancy outcome during the study period. The outcomes were determined using lists of clinical codes to denote the occurrence of either a delivery-related outcome (livebirth or stillbirth) or pregnancy loss (termination, miscarriage, ectopic pregnancy, molar pregnancy, probable termination, unspecified loss, or blighted ovum) using definitions and terminology used in previous studies[8,10], and also through consultation with the health practitioners listed above (Fig. 1—Stage 2 and 3).

### Structure of the pregnancy algorithm
**Defining a pregnancy outcome, episode, and conflict.** We defined pregnancy outcomes using terminology and definitions previously reported[8,10], namely, delivery or pregnancy loss. We identified the first pregnancy outcome in each woman's record during the study period using linked GP and HES records and the date on which this outcome occurred. We considered delivery outcomes that occurred at least 175 days (25 weeks) after a prior delivery outcome to indicate a subsequent pregnancy (i.e. a separate episode)[8]. Delivery records within 175 days of the previous episode were considered to be a reflection of the same pregnancy. Records of pregnancy loss that occurred within 56 days (8 weeks) of a previous record of a pregnancy loss outcome were considered the same episode. Pregnancy loss records without a clear outcome (e.g. ICD-10 code O06—Unspecified abortion, which may represent a miscarriage or termination) were categorised as terminations in accordance with the register's prioritisation process. The mean duration of a pregnancy with a full term delivery outcome was set at 275 days, which was determined using data from the most recent and complete year (2019) available in the Maternity Services Dataset 1[25]. This was used to estimate the duration of pregnancies with a livebirth outcome in the

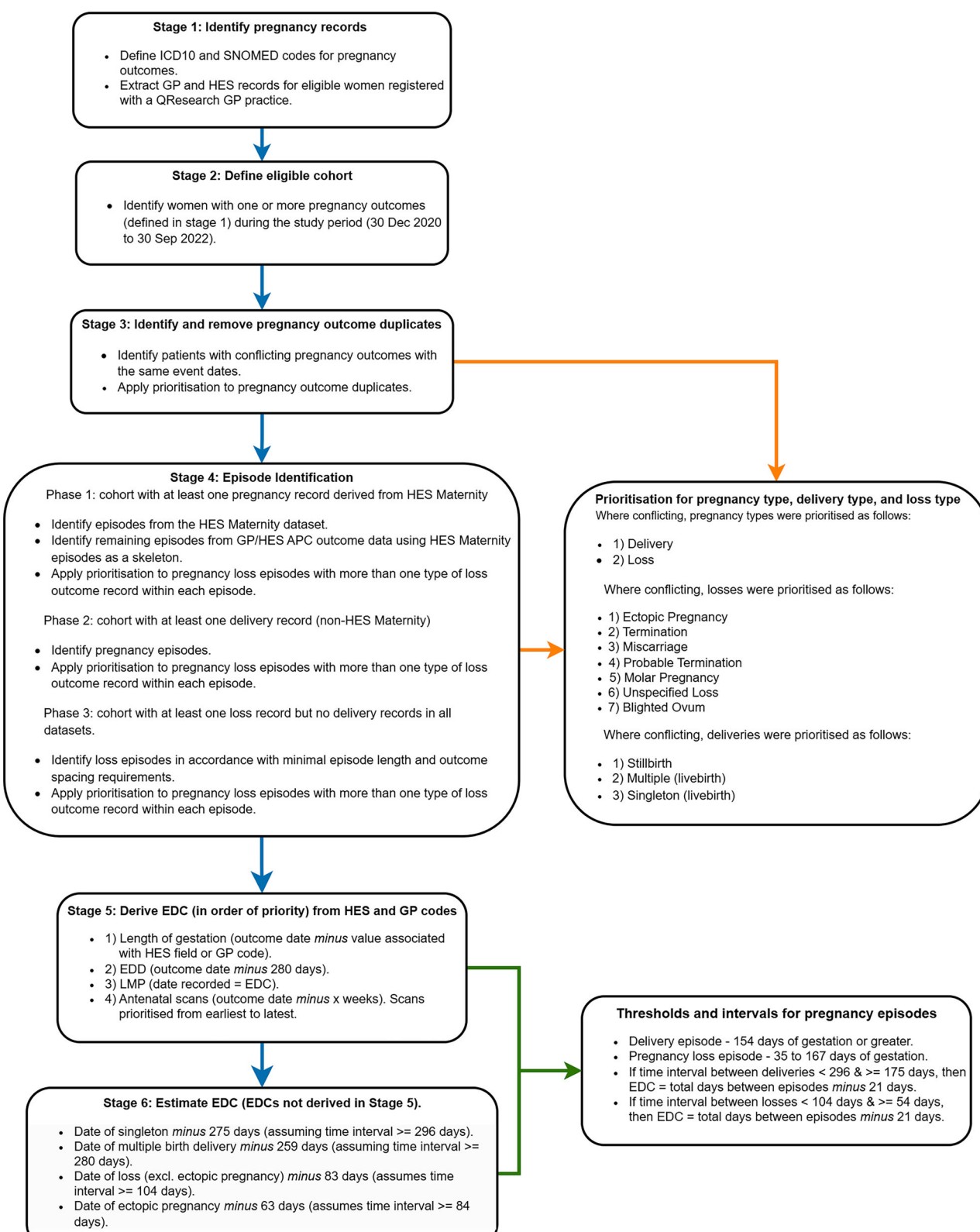

**Fig. 1 | Data flow diagram.** Diagram displaying the Stages (1–6) of the algorithm used for the construction of the pregnancy register. Text boxes separated by dark blue arrows = describe the purpose and the key steps within each Stage of the algorithm, as well as which subset of the pregnancy cohort is being processed, where applicable. Text box linked by orange arrows = describes the structure of the prioritisation process used for sorting pregnancy records with conflicting outcomes in Stage 3 and 4. Text box linked by green arrows = describes the thresholds and time intervals used to determine the start date (EDC) of each pregnancy, both derived (Stage 5) and estimated (Stage 6).

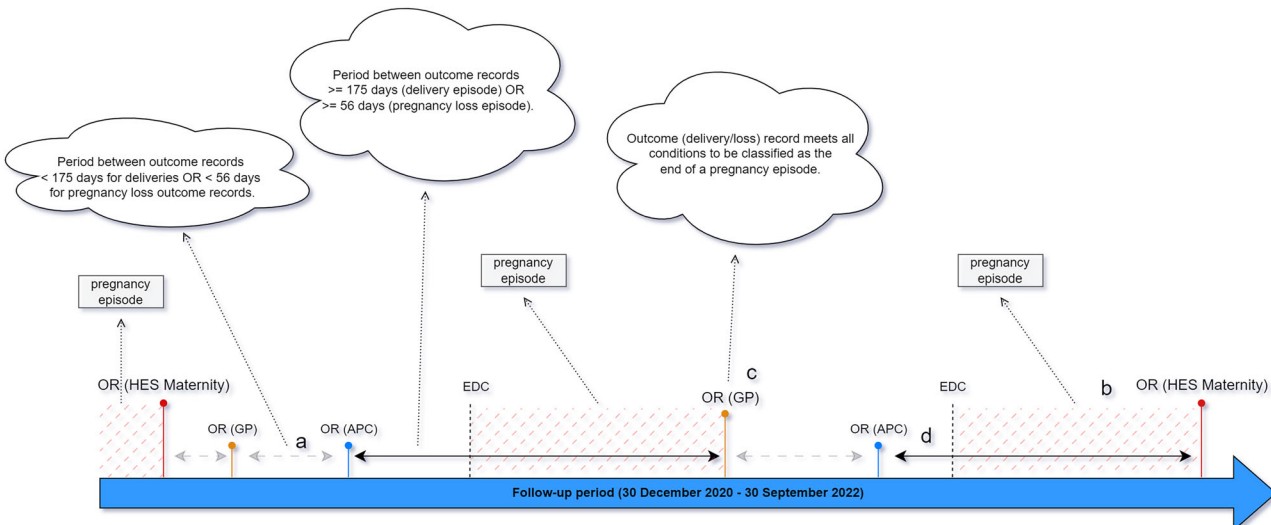

**Fig. 2 | Phase 1—Episode Identification.** Diagram displaying criteria used to identify qualifying outcome records (OR) from all datasets. **a** Period between outcome records is below minimum period required for creation of pregnancy episode —grey dashed line with arrows. **b** Pregnancy episode derived from HES Maternity dataset. **c** Pregnancy episode derived from qualifying database and from an outcome record which does not conflict with pre-existing episodes derived from HES Maternity (if any). **d** Period between outcome records is equal to or exceeds minimum period required for creation of pregnancy episode (delivery/loss)— solid black line with arrows. black dotted line = EDC (Estimated Date of Conception). Red line = outcome record from HES Maternity. Blue line = outcome record from HES Admissions. Orange line = outcome record from primary care. Red dashed lines = period between conception and pregnancy outcome. Large blue arrow = duration of study period.

absence of a recorded gestational age, as it was judged to be an accurate reflection of the length of the average, term singleton delivery. A code conflict was defined differently depending on the context and the type of record affected. They were:

- Outcome records of different type (delivery vs loss), but with the same date of recording. For example, a patient with a stillbirth and miscarriage recorded on the same day.
- Outcome records of the same type (both/all deliveries or losses), but with the same date of recording and different outcomes. For example, a patient with a termination and miscarriage recorded on the same day.
- Pregnancy loss records of different type (e.g. miscarriage, ectopic pregnancy, etc.), but which could be said to be a representation of the same episode (i.e. <56 days between at least 2 records, with at least 1 record occurring after another which is classified as a pregnancy loss episode by the register algorithm).

**Thresholds and parameters for estimating length of gestation.**

- All delivery episodes resulting in a livebirth were required to be more than 154 days in length, as this was considered to be the minimum time threshold for a viable birth[26] (singleton or multiple).
- Delivery episodes derived from the HES Maternity dataset with a recorded length of gestation of less than 154 days were dropped from the cohort.
- All delivery episodes resulting in a stillbirth were required to have at least 168 days of gestation[27]. The delivery of a non-living fetus prior to 168 days of gestation is classified as a miscarriage or late fetal loss[28,29].
- All pregnancy loss episodes were required to have between 35 and 167 days of gestation.
- A minimum of 21 days was required between the estimated delivery date of any previous pregnancy episode and first day of the mother's last menstrual period of any subsequent episode.
- The first event recorded for all women was assumed to be the date of their first episode (pregnancy outcome), with subsequent delivery and loss outcomes classified as episodes if there was a minimum of 175 and 56 days between events, respectively.

**Estimating the pregnancy end dates and identifying the pregnancy episodes (Fig. 1—Stage 4).** The pregnancy algorithm processed all available pregnancy and delivery events which could be linked to the eligible cohort. Pregnancy episodes were identified in three phases with each phase processing a different subset of the cohort.

**Phase 1: define cohort with at least one pregnancy record derived from HES Maternity records.** The first phase defined the pregnancy episodes and end dates for deliveries and pregnancy losses in the HES Maternity dataset. These episodes formed the "spine" of the register and were prioritised over events recorded in the GP and other HES data, where applicable and conflicting. All HES Maternity records were classified as episodes, providing the time elapsed between pregnancy outcomes did not conflict with the weeks of gestation recorded for each pregnancy for women with multiple pregnancy outcomes recorded, and the time interval between outcome records (one outcome record per row) was greater than the required thresholds for each pregnancy outcome. In the event of the length of gestation not being recorded, pregnancies received an estimated value (simple imputation), which was determined and constrained by both the pregnancy outcome and the time interval between each pregnancy episode. A visual representation of phase 1 is shown in Fig. 2.

*Example*: 83 days of gestation was imputed for a pregnancy loss if there was no record of a previous pregnancy event at least 104 (83 + 21) days prior to the loss outcome date. 21 days was considered to be the minimum, acceptable length of time between the end of one pregnancy and the start of another for any individual. If the period of available time was less than 104 days, the length of gestation was calculated as the available time minus 21 days. However, the estimated date of conception could not occur less than 35 days prior to the loss outcome date.

A pre-determined prioritisation using new (to the best of our knowledge) and previously reported methods[8] was applied to all delivery/loss events derived from all data sources where an individual had multiple but different outcomes of the same or differing type (delivery/loss). This prioritisation was applied to all conflicting outcomes (deliveries & losses) with the same date of recording and conflicting outcomes (losses only)

which fell within the same episode. Deliveries were prioritised over losses where outcomes were recorded on the same date. Individuals with multiple and/or conflicting delivery-related GP (SNOMED CT) or ICD-10 codes with the same date of recording were deduplicated and the excess codes removed from the data. Conflicting delivery codes were retained in the following order of priority; 1) stillbirths; 2) livebirth (multiples); 3) livebirth (singleton). To ensure that a pregnancy outcome code represented a unique pregnancy episode and was not representative of a historical pregnancy, a minimum of 175 days was required following any previous pregnancy outcome for a delivery outcome, and a minimum of 56 days for a pregnancy loss. Where there were multiple and conflicting pregnancy loss codes, either with the same date or within the same episode, they were prioritised as follows: 1) ectopic pregnancy; 2) termination; 3) miscarriage; 4) probable termination of pregnancy; 5) molar pregnancy; 6) unspecified loss; and 7) blighted ovum[8]. Conflicting pregnancy loss codes were dropped from the register after the prioritisation was complete.

**Phase 2: define cohort with at least one delivery record (non-HES Maternity).** The second phase of the algorithm processed all pregnancy outcomes for individuals without an associated record in the HES Maternity dataset using codes in primary care records, OPCS, and HES APC only. Minimum time intervals of 175 and 56 days were required between delivery and pregnancy loss outcome events, respectively, for an outcome event to be classified as the end of an episode, as previously described in phase 1. The pregnancy loss prioritisation algorithm[8] (as used in phase 1) was applied to all individuals identified in the phase 2 subset. A visual representation of phase 2 is shown in Supp. Figure 1.

**Phase 3: define sub-set cohort with at least one pregnancy loss record but no delivery records in all datasets.** The third and last phase defined the pregnancy loss outcomes associated with individuals with no delivery records in any of the datasets. A minimum time interval of 56 days was required between pregnancy loss outcomes for events to be classified as the end of an episode. The pregnancy loss prioritisation algorithm[8], as used in phase 1 and 2, was applied to all individuals identified in the phase 3 subset. A visual representation of phase 3 is shown in Supp. Fig. 2.

**Defining the estimated date of conception (Fig. 1—Stage 5 and 6).** We used methods described in previous studies[8,30,31] to estimate the date of conception, using these variables in the following order of priority according to data availability:

1) Length of gestation (HES field or GP code): estimated date of conception = delivery date *minus* Length of gestation in days. (Preference was given to HES if both were present). The HES field (GESTAT_n) was converted from whole weeks to days (weeks of gestation x 7).
2) Estimated Date of Delivery (EDD) reported in GP records: estimated date of conception = EDD *minus* 280 days.*
3) Date of last menstrual period—first day (LMP) (GP code): estimated date of conception = LMP.
4) Antenatal scans—scan date *minus* associated weekly value (number of completed weeks) from date of scan (with preference towards earlier scans and those recorded in HES over GP records).
5) If none of the above were recorded, we estimated (simple imputation) the date of conception as:
   - Date of full-term delivery *minus* 275 days assuming a minimum of 296 days between pregnancy episodes.
   - Date of pre-term delivery *minus* 259 days assuming a minimum of 280 days between pregnancy episodes.**
   - Date of multiple birth delivery *minus* 259 days assuming a minimum of 280 days between pregnancy episodes.
   - Date of early pregnancy loss outcome (excluding ectopic pregnancy) *minus* 83 days assuming a minimum of 104 days between pregnancy episodes.

**Table 1 | Demographic characteristics of women who had one or more pregnancies during the COVID-19 Vaccination Program (30 Dec 2020–30 Sept 2022)**

| Total number in cohort | 266,758 |
|---|---|
| Age in 2020, mean (SD) | 30 (5.7) |
| Age group, *N* (column %) | |
| <20 | 10,573 (4.0%) |
| 20–29 | 110,787 (41.5%) |
| 30–34 | 87,279 (32.7%) |
| 35–39 | 46,662 (17.5%) |
| =>40 | 11,457 (4.3%) |
| Ethnic group, *N* (column %) | |
| White | 191,951 (72.0%) |
| Indian | 11,319 (4.2%) |
| Pakistani | 10,735 (4.0%) |
| Bangladeshi | 7368 (2.8%) |
| Other Asian | 5879 (2.2%) |
| Caribbean | 3230 (1.2%) |
| Black African | 11,578 (4.3%) |
| Chinese | 2149 (0.8%) |
| Other | 16,714 (6.3%) |
| Missing | 5835 (2.2%) |
| Quintile of Townsend, *N* (column %) | |
| 1—most affluent | 43,596 (16.3%) |
| 2 | 51,026 (19.1%) |
| 3 | 55,697 (20.9%) |
| 4 | 55,634 (20.9%) |
| 5—most deprived | 52,614 (19.7%) |
| Not recorded | 8191 (3.1%) |
| Region, *N* (column %) | |
| East Midlands | 4510 (1.7%) |
| East of England | 9860 (3.7%) |
| London | 80,933 (30.3%) |
| North East | 5698 (2.1%) |
| North West | 49,053 (18.4%) |
| South Central | 29,824 (11.2%) |
| South East | 29,811 (11.2%) |
| South West | 21,563 (8.1%) |
| West Midlands | 28,311 (10.6%) |
| Yorkshire & Humber | 7195 (2.7%) |

- Date of ectopic pregnancy outcome *minus* 63 days assuming a minimum of 84 days between pregnancy episodes.

*The EDD is calculated in general practice at the beginning of a pregnancy by adding 280 days to the first day of the mother's last menstrual period.*
**Only deliveries with a qualifying ICD10 code (0601 or 0603) which were derived from HES Maternity.*

**Presentation of results**
We described the demographic characteristics of all women who were included in the register, defined as having one or more pregnancies during the 21-month study period (Table 1). This included age at start of 2020, self-assigned ethnic group (white, Indian, Pakistani, Bangladeshi, Other Asian, Caribbean, Black African, Chinese, Other), quintile of social deprivation, and geographical region of England of

**Table 2 | Demographic characteristics of cohort by number of pregnancies during study period (30 December 2020–30 September 2022)**

| Number of pregnancies | | | | |
|---|---|---|---|---|
| | **1** | **2** | **3 or more** | **Total** |
| *N* (row %) | 266,758 (95.6%) | 11,690 (4.2%) | 579 (0.2%) | 279,027 |
| Age at outcome (delivery/loss) date, mean (SD) | 31 (5.7) | 31 (5.8) | 32 (5.9) | 31 (5.7) |
| Age group, *N* (column %) | | | | |
| 15–19 | 5246 (2.0%) | [149–152][a] | <5[b] | 5399 (1.9%) |
| 20–24 | 29,917 (11.2%) | 1421 (12.2%) | 58 (10.0%) | 31,396 (11.3%) |
| 25–29 | 63,057 (23.6%) | 2677 (22.9%) | 119 (20.6%) | 65,853 (23.6%) |
| 30–34 | 89,703 (33.6%) | 3631 (31.1%) | 162 (28.0%) | 93,496 (33.5%) |
| 35–39 | 59,550 (22.3%) | 2762 (23.6%) | 160 (27.6%) | 62,472 (22.4%) |
| 40–44 | 17,655 (6.6%) | 994 (8.5%) | 74 (12.8%) | 18,723 (6.7%) |
| 45–50 | 1630 (0.6%) | [54–57][a] | <5[b] | 1688 (0.6%) |
| Ethnic group, *N* (row %) | | | | |
| White | 191,951 (95.5%) | 8555 (4.3%) | 447 (0.2%) | 200,953 |
| Indian | 11,319 (96.3%) | 416 (3.5%) | 22 (0.2%) | 11,757 |
| Pakistani | 10,735 (94.6%) | 583 (5.1%) | 24 (0.2%) | 11,342 |
| Bangladeshi | 7368 (95.1%) | 363 (4.7%) | 18 (0.2%) | 7749 |
| Other Asian | 5879 (96.7%) | 193 (3.2%) | 10 (0.2%) | 6082 |
| Caribbean | 3230 (95.6%) | 141 (4.2%) | 6 (0.2%) | 3377 |
| Black African | 11,578 (95.8%) | 485 (4.0%) | 23 (0.2%) | 12,086 |
| Chinese | 2149 (97.0%) | [62–65][a] | <5[b] | 2215 |
| Other | 16,714 (96.2%) | 636 (3.7%) | 17 (0.1%) | 17,367 |
| Not Recorded | 5,835 (95.7%) | 255 (4.2%) | 9 (0.1%) | 6099 |
| Quintile of Townsend, *N* (row %) | | | | |
| 1—most affluent | 43,596 (95.6%) | 1894 (4.2%) | 124 (0.3%) | 45,614 |
| 2 | 51,026 (95.6%) | 2199 (4.1%) | 132 (0.2%) | 53,357 |
| 3 | 55,697 (95.7%) | 2420 (4.2%) | 100 (0.2%) | 58,217 |
| 4 | 55,634 (95.7%) | 2437 (4.2%) | 92 (0.2%) | 58,163 |
| 5—most deprived | 52,614 (95.5%) | 2370 (4.3%) | 111 (0.2%) | 55,095 |
| Not recorded | 8191 (95.5%) | 370 (4.3%) | 20 (0.2%) | 8581 |
| Region, *N* (row %) | | | | |
| East Midlands | 4510 (95.1%) | 213 (4.5%) | 19 (0.4%) | 4742 |
| East of England | 9860 (95.7%) | 422 (4.1%) | 17 (0.2%) | 10,299 |
| London | 80,933 (95.9%) | 3279 (3.9%) | 142 (0.2%) | 84,354 |
| North East | 5698 (95.3%) | 270 (4.5%) | 11 (0.2%) | 5979 |
| North West | 49,053 (95.3%) | 2305 (4.5%) | 125 (0.2%) | 51,483 |
| South Central | 29,824 (95.5%) | 1317 (4.2%) | 72 (0.2%) | 31,213 |
| South East | 29,811 (96.1%) | 1152 (3.7%) | 58 (0.2%) | 31,021 |
| South West | 21,563 (95.1%) | 1054 (4.6%) | 68 (0.3%) | 22,685 |
| West Midlands | 28,311 (95.3%) | 1354 (4.6%) | 49 (0.2%) | 29,714 |
| Yorkshire & Humber | 7195 (95.5%) | 324 (4.3%) | 18 (0.2%) | 7537 |

[a]Count obscured to prevent deductive disclosure of censored counts (< 5).
[b]Censored count does not include 0.

the women. Ethnicity data were derived from patients' GP records where available and supplemented by HES ethnicity data when missing from patients' GP records. All other descriptive analyses (Tables 2 to 5) were carried out at the pregnancy level using age at the date of delivery or pregnancy loss, where applicable, with the same demographic factors used in Table 1. We described the demographics by the number of pregnancies recorded during follow-up (Table 2) and the data sources used to define the pregnancies (Supplementary Table 1). We described the demographic characteristics for all pregnancies, stratified by pregnancy outcome (delivery or pregnancy loss—Table 3). For deliveries, we described the demographic factors by delivery outcome (livebirth or stillbirth—Table 4). For pregnancy losses (Table 5), we stratified by miscarriage, termination of pregnancy or other pregnancy losses (including ectopic, molar pregnancy, and blighted ovum).

As the temporal coverage of the HES and GP data used in the construction of the pregnancy register differed in the context of this study, it has the potential to impact the representation of certain pregnancy outcomes more than others (e.g. pregnancy losses, specifically miscarriages, are more likely to be derived from primary rather than secondary care records).

**Table 3 | Demographic characteristics of cohort pregnancies by outcome (Delivery or Loss) during study period (30 December 2020–30 September 2022)**

| Pregnancy loss or delivery outcome | | | |
|---|---|---|---|
| | Delivery (livebirth or stillbirth) | Loss | Total |
| Number of pregnancies (row %) | 232,673 (83.4%) | 46,354 (16.6%) | 279,027 |
| Age at outcome (delivery/loss) date, mean (SD) | 31 (5.5) | 31 (6.9) | 31 (5.7) |
| Age group, N (row %) | | | |
| <20 | 3646 (67.5%) | 1753 (32.5%) | 5399 |
| 20–29 | 81,772 (84.1%) | 15,477 (15.9%) | 97,249 |
| 30–34 | 81,345 (87.0%) | 12,151 (13.0%) | 93,496 |
| 35–39 | 52,004 (83.2%) | 10,468 (16.8%) | 62,472 |
| =>40 | 13,906 (68.1%) | 6505 (31.9%) | 20,411 |
| Ethnic group, N (row %) | | | |
| White | 167,300 (83.3%) | 33,653 (16.7%) | 200,953 |
| Indian | 9878 (84.0%) | 1879 (16.0%) | 11,757 |
| Pakistani | 9763 (86.1%) | 1579 (13.9%) | 11,342 |
| Bangladeshi | 6636 (85.6%) | 1113 (14.4%) | 7749 |
| Other Asian | 5209 (85.6%) | 873 (14.4%) | 6082 |
| Caribbean | 2596 (76.9%) | 781 (23.1%) | 3377 |
| Black African | 10,020 (82.9%) | 2066 (17.1%) | 12,086 |
| Chinese | 1847 (83.4%) | 368 (16.6%) | 2215 |
| Other | 14,525 (83.6%) | 2842 (16.4%) | 17,367 |
| Not recorded | 4899 (80.3%) | 1200 (19.7%) | 6099 |
| Quintile of Townsend, N (row %) | | | |
| 1—most affluent | 38,098 (83.5%) | 7516 (16.5%) | 45,614 |
| 2 | 44,574 (83.5%) | 8783 (16.5%) | 53,357 |
| 3 | 48,597 (83.5%) | 9620 (16.5%) | 58,217 |
| 4 | 48,505 (83.4%) | 9658 (16.6%) | 58,163 |
| 5—most deprived | 45,520 (82.6%) | 9575 (17.4%) | 55,095 |
| Not Recorded | 7379 (86.0%) | 1202 (14.0%) | 8581 |
| Region, N (row %) | | | |
| East Midlands | 3771 (79.5%) | 971 (20.5%) | 4742 |
| East of England | 8818 (85.6%) | 1481 (14.4%) | 10,299 |
| London | 71,457 (84.7%) | 12,897 (15.3%) | 84,354 |
| North East | 4688 (78.4%) | 1291 (21.6%) | 5979 |
| North West | 42,310 (82.2%) | 9173 (17.8%) | 51,483 |
| South Central | 26,069 (83.5%) | 5144 (16.5%) | 31,213 |
| South East | 26,464 (85.3%) | 4557 (14.7%) | 31,021 |
| South West | 18,118 (79.9%) | 4567 (20.1%) | 22,685 |
| West Midlands | 24,784 (83.4%) | 4930 (16.6%) | 29,714 |
| Yorkshire & Humber | 6194 (82.2%) | 1343 (17.8%) | 7537 |

Supplementary analyses were conducted on a restricted cohort of women and pregnancies, which were identified by the algorithm during the study period up to and including 11 February 2022. This will allow the impact of the distribution of pregnancy outcomes generated by the register algorithm to be investigated. These supplementary analyses were performed at the patient (Supplementary Data 4) and pregnancy level (Supplementary Data 5–9). All the main tables were produced and the data management/analyses conducted using Stata MP, version 18.

**Validation**

**Internal validation.** To support the validity of the pregnancy episodes identified and characterised by the algorithm, the four contributing datasets were checked for evidence of pregnancy outcome codes of the same type (e.g. delivery or loss) recorded on or around the pregnancy outcome date of each episode. The datasets were searched for corresponding records (same type) occurring on the same day as the outcome date; within 2 days (±) of the outcome date; and within 7 days (±) of the outcome date of each pregnancy episode. A higher level of agreement between data sources supports the accuracy of the timing of each episode, the completeness of the dataset, and the sensitivity of the algorithm. Pregnancy outcomes were considered to be "linked/corresponding events" if they fell into one of the time frames, were of the same type, and did not represent the end of an episode as defined by the algorithm. The Positive Predictive Value (PPV), defined as the proportion of episodes with linked records in at least one other database, was calculated for deliveries and losses and for each period independently. The formula

**Table 4 | Demographic characteristics of cohort deliveries by outcome during study period (30 December 2020–30 September 2022)**

| Outcome associated with delivery record | | | |
|---|---|---|---|
| | **Stillbirth** | **Livebirth** | **Total** |
| Number of deliveries (row %) | 1,054 (0.5%) | 231,619 (99.5%) | 232,673 |
| Age at delivery date, mean (SD) | 31 (6.0) | 31 (5.5) | 31 (5.5) |
| Age group, N (row %) | | | |
| <20 | 20 (0.5%) | 3626 (99.5%) | 3646 |
| 20–29 | 374 (0.5%) | 81,398 (99.5%) | 81,772 |
| 30–34 | 327 (0.4%) | 81,018 (99.6%) | 81,345 |
| 35–39 | 256 (0.5%) | 51,748 (99.5%) | 52,004 |
| =>40 | 77 (0.6%) | 13,829 (99.4%) | 13,906 |
| Ethnic group, N (row %) | | | |
| White | 684 (0.4%) | 166,616 (99.6%) | 167,300 |
| Indian | 49 (0.5%) | 9829 (99.5%) | 9878 |
| Pakistani | 54 (0.6%) | 9709 (99.4%) | 9763 |
| Bangladeshi | 35 (0.5%) | 6601 (99.5%) | 6636 |
| Other Asian | 38 (0.7%) | 5171 (99.3%) | 5209 |
| Caribbean | 19 (0.7%) | 2577 (99.3%) | 2596 |
| Black African | 72 (0.7%) | 9948 (99.3%) | 10,020 |
| Chinese | 5 (0.3%) | 1842 (99.7%) | 1847 |
| Other | 69 (0.5%) | 14,456 (99.5%) | 14,525 |
| Not recorded | 29 (0.6%) | 4870 (99.4%) | 4899 |
| Quintile of Townsend, N (row %) | | | |
| 1—most affluent | 148 (0.4%) | 37,950 (99.6%) | 38,098 |
| 2 | 176 (0.4%) | 44,398 (99.6%) | 44,574 |
| 3 | 214 (0.4%) | 48,383 (99.6%) | 48,597 |
| 4 | 227 (0.5%) | 48,278 (99.5%) | 48,505 |
| 5—most deprived | 268 (0.6%) | 45,252 (99.4%) | 45,520 |
| Not recorded | 21 (0.3%) | 7358 (99.7%) | 7379 |
| Region, N (row %) | | | |
| East Midlands | 15 (0.4%) | 3756 (99.6%) | 3771 |
| East of England | 30 (0.3%) | 8788 (99.7%) | 8818 |
| London | 345 (0.5%) | 71,112 (99.5%) | 71,457 |
| North East | 19 (0.4%) | 4669 (99.6%) | 4688 |
| North West | 185 (0.4%) | 42,125 (99.6%) | 42,310 |
| South Central | 115 (0.4%) | 25,954 (99.6%) | 26,069 |
| South East | 117 (0.4%) | 26,347 (99.6%) | 26,464 |
| South West | 67 (0.4%) | 18,051 (99.6%) | 18,118 |
| West Midlands | 121 (0.5%) | 24,663 (99.5%) | 24,784 |
| Yorkshire & Humber | 40 (0.6%) | 6154 (99.4%) | 6194 |

(Eq. 1) used to calculate the PPV is shown below:

$$PPV = \left( \frac{Total\ number\ of\ linked\ records}{Total\ number\ of\ records\ in\ register} \right) \times 100 \quad (1)$$

All delivery records were also checked for linked postnatal GP records within 175 days of the delivery date. The level of agreements/overlap between datasets for deliveries and losses were presented in Venn Diagrams generated using the Eulerr package in RStudio (version 2023.12.1 + 402 "Ocean Storm")[32]. As part of the supplementary analyses, the internal validation was also conducted on all pregnancies which concluded during the study period up to or before 4 February 2022 (seven days prior to the end

of the available primary care data to ensure that data were available for linking). This was done to assess the impact of the restricted dates for primary care data during the study period on the PPV and the level of agreement between datasets. The internal validation does not describe the likelihood of identifying linked records in the available data sources by the data sources used to identify the final episodes. However, it is possible to deduce which data source were used (in some cases) to identify episodes using the register's prioritisation process.

**External validation.** Delivery (livebirth, stillbirth) and pregnancy loss rates (miscarriage, termination, ectopic pregnancy, molar pregnancy, blighted ovum) in the register were compared to national (Office for National Statistics [ONS] and the UK Department of Health) and other published statistics[33,34]. Where applicable and stated, register data from 2021 were selected for comparison to national statistics as this was the only complete calendar year included in the study period. A restricted age range (15 to 44 years) was applied to the pregnancy register when comparing against published statistics as this is a common convention for the calculation of fertility rates as it represents the main childbearing years[35]. While pregnancies do occur amongst those outside of this age range, the relatively small number of cases relative to the size of populations dilutes the rate fraction. The numerators used in the calculation of the QResearch rates were restricted to the 15 to 44 age range and were directly age-standardised when the denominator included all eligible women in the database. The standard population was based on mid-year estimates for the female population of England between the ages of 15 to 44 in 2021 published by ONS[36].

### Patient and public involvement
As a project team, we established a patient and public involvement and engagement (PPIE) panel including participants from the Centre for Ethnic Health Research with lived experience of pregnancy during the COVID-19 Vaccination Programme. They helped us develop our proposal.

### Ethics
QResearch is a Research Ethics Approved Research Database with ongoing approval from the East Midlands Multi-Centre Research Ethics Committee (Ref: 18/EM/0400). This study was approved by the QResearch Scientific Committee on 9th June 2022. This research protocol[20] has been developed with support from a patient and public involvement panel, who will continue to provide input throughout the duration of the study. Participant GP data is provided by Optum through the upload of anonymised, patient data from participating GP practices using EMIS Web across the UK. Practices may withdraw their participation at any time without providing a reason. As the patient data is anonymised, informed consent is obtained and required from the guardian (GP practice) of the records. Participating GP practices are required to inform their patients of the practice's participation, and patients at these practices may withdraw their participation through the National Data Opt-Out service at any time. Further information may be found on the QResearch website: https://www.qresearch.org/about/ethics-and-confidentiality/.

### Results
The cohort consisted of 266,758 women with 279,027 pregnancy outcomes recorded during the study period. Table 1 shows the characteristics of women included in the cohort. The majority (74.2%) of women were between 20-34 years of age, with the largest proportion (41.5%) of pregnancies occurring among those between 20-29 years. The cohort had a mean age of 30 years (Standard Deviation (SD) = 5.7) at study start. 72% were white, 4.3% Black African, 4.2% Indian, 4.0% Pakistani, 2.8% Bangladeshi, 2.2% Other Asian, 1.2% Caribbean, 0.8% Chinese, 6.3% were defined as "other" ethnicity and 2.2% had missing ethnicity data. The supplementary analyses in the restricted cohort (n = 190,467) of women with pregnancy outcomes up to 11 February 2022 showed comparable results, with the majority (68.5%) of women between 20-34 years of age with a mean age of

**Table 5 | Demographic Characteristics of cohort pregnancy losses by outcome during study period (30 December 2020–30 September 2022)**

| Type of pregnancy loss | | | | |
|---|---|---|---|---|
| | **Miscarriage** | **Termination** | **Other pregnancy losses** | **Total** |
| Number of pregnancy losses (row %) | 19,648 (42.4%) | 22,655 (48.9%) | 4,051 (8.7%) | 46,354 |
| Age at loss date, mean (SD) | 32 (6.5) | 31 (7.2) | 32 (6.1) | 31 (6.9) |
| Age group, N (row %) | | | | |
| <20 | 500 (28.5%) | 1179 (67.3%) | 74 (4.2%) | 1753 |
| 20–29 | 5854 (37.8%) | 8268 (53.4%) | 1355 (8.8%) | 15,477 |
| 30–34 | 5539 (45.6%) | 5403 (44.5%) | 1209 (9.9%) | 12,151 |
| 35–39 | 4763 (45.5%) | 4730 (45.2%) | 975 (9.3%) | 10,468 |
| =>40 | 2992 (46.0%) | 3075 (47.3%) | 438 (6.7%) | 6505 |
| Ethnic group, N (row %) | | | | |
| White | 13,877 (41.2%) | 16,949 (50.4%) | 2827 (8.4%) | 33,653 |
| Indian | 918 (48.9%) | 799 (42.5%) | 162 (8.6%) | 1879 |
| Pakistani | 904 (57.3%) | 524 (33.2%) | 151 (9.6%) | 1579 |
| Bangladeshi | 603 (54.2%) | 414 (37.2%) | 96 (8.6%) | 1113 |
| Other Asian | 424 (48.6%) | 374 (42.8%) | 75 (8.6%) | 873 |
| Caribbean | 281 (36.0%) | 384 (49.2%) | 116 (14.9%) | 781 |
| Black African | 882 (42.7%) | 993 (48.1%) | 191 (9.2%) | 2066 |
| Chinese | 149 (40.5%) | 188 (51.1%) | 31 (8.4%) | 368 |
| Other | 1188 (41.8%) | 1361 (47.9%) | 293 (10.3%) | 2842 |
| Not recorded | 422 (35.2%) | 669 (55.8%) | 109 (9.1%) | 1200 |
| Quintile of Townsend, N (row %) | | | | |
| 1—most affluent | 3318 (44.1%) | 3571 (47.5%) | 627 (8.3%) | 7516 |
| 2 | 3756 (42.8%) | 4303 (49.0%) | 724 (8.2%) | 8783 |
| 3 | 4112 (42.7%) | 4652 (48.4%) | 856 (8.9%) | 9620 |
| 4 | 4045 (41.9%) | 4760 (49.3%) | 853 (8.8%) | 9658 |
| 5—most deprived | 3859 (40.3%) | 4831 (50.5%) | 885 (9.2%) | 9575 |
| Not recorded | 558 (46.4%) | 538 (44.8%) | 106 (8.8%) | 1202 |
| Region, N (row %) | | | | |
| East Midlands | 360 (37.1%) | 548 (56.4%) | 63 (6.5%) | 971 |
| East of England | 706 (47.7%) | 635 (42.9%) | 140 (9.5%) | 1481 |
| London | 5130 (39.8%) | 6452 (50.0%) | 1315 (10.2%) | 12,897 |
| North East | 410 (31.8%) | 802 (62.1%) | 79 (6.1%) | 1291 |
| North West | 4182 (45.6%) | 4225 (46.1%) | 766 (8.4%) | 9173 |
| South Central | 2369 (46.1%) | 2302 (44.8%) | 473 (9.2%) | 5144 |
| South East | 1928 (42.3%) | 2193 (48.1%) | 436 (9.6%) | 4557 |
| South West | 1761 (38.6%) | 2,477 (54.2%) | 329 (7.2%) | 4567 |
| West Midlands | 2260 (45.8%) | 2309 (46.8%) | 361 (7.3%) | 4930 |
| Yorkshire & Humber | 542 (40.4%) | 712 (53.0%) | 89 (6.6%) | 1343 |

30 (SD = 5.8). Similar levels of representation were observed across all ethnicities (Supplementary Data 4).

Table 2 shows population characteristics by number of pregnancies: 266,758 women had a single pregnancy during follow-up, 11,690 had two pregnancies, 553 had three pregnancies, and 26 had four pregnancies. The proportion of women with two pregnancies during the study period was between 3–4.5% across all ethnic groups with the exception of those of Pakistani (5.1%), Bangladeshi (4.7%), and Chinese (2.8%) ethnicity. Otherwise, patterns were similar across ethnic, deprivation, and regional groups. Comparable results were seen in the supplementary analyses of the restricted cohort (Supplementary Data 5), although the proportion of women with two pregnancies ranged from 1.4–2.2% across ethnic groups, which is to be expected given the shortened observation period.

Demographic characteristics of the cohort, stratified by pregnancy outcome (delivery or loss) are shown in Table 3. Of all the pregnancies, 232,673 (83.4%) resulted in a delivery and 46,354 (16.6%) ended in a pregnancy loss. The mean age of women at date of delivery was 31 years (SD 5.5) and 31 years (SD 6.9) at date of pregnancy loss. The proportion of women recorded as having a delivery compared to those who experienced a loss, varied by ethnicity and geographic region. Women of Pakistani (86.1%), Bangladeshi (85.6%), and Other Asian (85.6%) ethnicity had the highest proportions of deliveries recorded, with the lowest proportion of deliveries and, consequently, the highest proportion of losses recorded amongst women of Caribbean (deliveries—76.9%; losses—23.1%), Black African (deliveries—82.9%; losses—17.1%) ethnicity, as well as those with no recorded ethnicity (deliveries—80.3%; losses—19.7%). There were also variations in pregnancy outcomes by region, with the highest proportion of deliveries observed amongst women in the East of England (85.6%) and the lowest in the north-east of England (78.4%). The proportion of pregnancies resulting in a delivery was also lowest in the 40-50 age group (62.5%) and the 15 to 20 age group (67.5%) and highest in the 30-34 age group (87%). The supplementary analyses (Supplementary Data 6) produced comparable results with a larger proportion of pregnancies resulting in a loss (19.5%). This increase is represented fairly equally across all demographic factors, with a 2–4% increase in the proportion of losses across all ethnic groups, regions, age groups, and levels of deprivation compared to the main analyses.

231,619 deliveries resulted in a livebirth (99.5%) and 1,054 in stillbirths (0.5%). Characteristics of women with a delivery outcome are shown in Table 4. Proportions of stillbirths were similar across all age groups and quintiles of deprivation, with the highest proportion among those in the oldest (40 to 50; 0.6%) and most deprived (Quintile 5; 0.6%) groups. The largest proportions of stillbirths were amongst those of Caribbean (0.7%), Black African (0.7%), and Other Asian (0.7%) ethnicity, with the lowest in those of Chinese (0.3%) and White (0.4%) ethnicity. Yorkshire & Humber (0.6%) was the region with the highest proportion of stillbirths compared to the East of England (0.3%), which had the lowest. The supplementary analyses (Supplementary Data 7) produced very similar results, with identical proportions of deliveries resulting in a livebirth (99.5%) and stillbirth (0.5%) compared to the main analyses.

Characteristics of the populations with different pregnancy loss outcomes are shown in Table 5. Pregnancy losses consisted of 19,648 (42.4%) miscarriages, 22,655 (48.9%) terminations, and 4,051 (8.7%) other pregnancy losses (ectopic pregnancies, molar pregnancies, and blighted ovum cases). The highest proportion of terminations (67.3%) and the lowest proportion of miscarriages (28.5%) were found in the 15-20 age group. The highest proportion of miscarriages was observed in those in the 40-50 age group. In terms of ethnicity, the highest proportions of miscarriages were amongst those of Pakistani (57.3%) and Bangladeshi (54.2%) ethnicity, and lowest in those of Caribbean (37.1%) ethnicity. The proportion of miscarriages was higher than the proportion of terminations in those with missing deprivation data (46.4% vs 44.8%) and in the East of England (47.7% vs 42.9%). All other deprivation groups and regions of England had higher proportions of terminations compared to miscarriages. The highest proportions of terminations were amongst those of White (50.4%), Chinese (51.1%), Caribbean (49.2%), and Black African (48.1%) ethnicity. High proportions were also present for those of Other (47.9%) and no recorded ethnicity (55.8%), as well as those in the North-East (62.1%) region compared to other parts of England. The proportion of pregnancy losses which were neither a miscarriage nor termination (other pregnancy losses), was similar by ethnic group (8.4% to 10.3%) apart from those of Caribbean

**Fig. 3 | Venn diagram—evidence of linking delivery records across ($n = 232,673$)\* data sources (±2 days of outcome date)—30 December 2020 to 30 September 2022.** \*Pregnancies suppressed by the Eulerr package due to low counts or omitted because of space restrictions: GP & HES Procedures ($n = 187$). Pink circle = HES Procedures; blue circle = HES Maternity; orange circle = GP. Black arrow = indicates section of diagram described by neighbouring label. Overlapping areas contain pregnancies with linking records in 1 or more datasets.

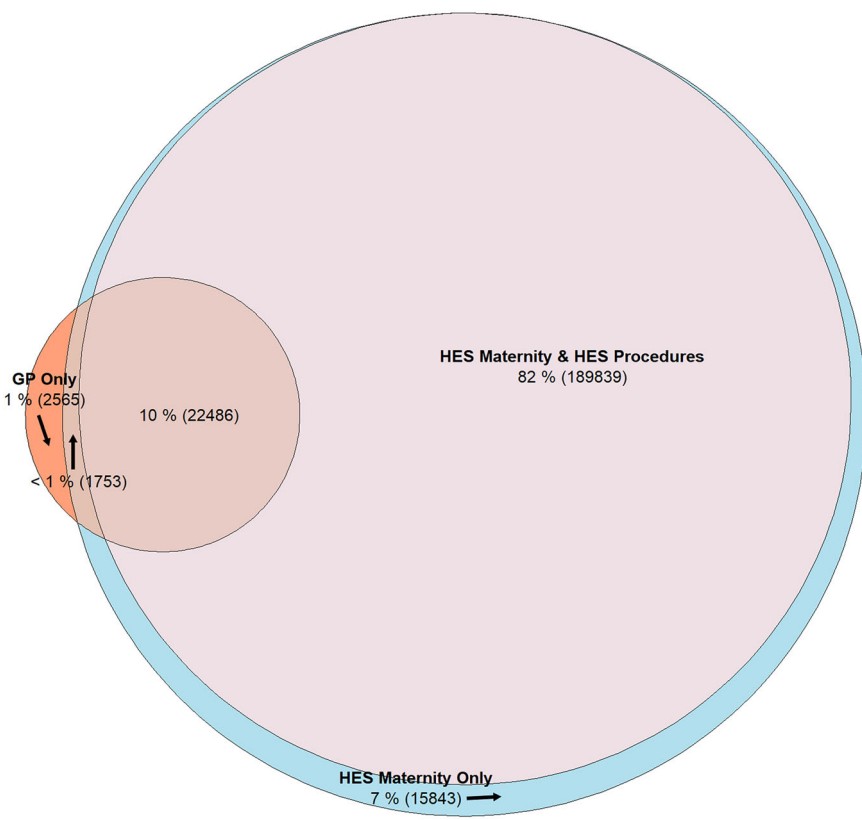

(14.9%) ethnicity. The supplementary analyses (Supplementary Data 8) produced comparable results. However, the highest proportion of pregnancy losses were miscarriages (47%) with a marginally lower proportion of terminations (45%). Other pregnancy losses were similar to the main analyses with a proportion of 8%. The proportion of miscarriages was similar by ethnic group with the highest proportion remaining in those of Pakistani ethnicity (62.1%). The distribution of miscarriages and terminations were similar across all other covariates (age group, level of deprivation, and region) with a 3-5% increase in miscarriages and a commensurate drop in the proportion of terminations.

Most pregnancies were derived from the HES Maternity (82.4%) dataset with an additional 10.6% from GP records, 3.4% from HES APC, and 3.6% from HES OPCS datasets. Almost all delivery episodes (98.8%) were identified from the HES Maternity dataset, whereas the majority of pregnancy losses (58.1% - including terminations) came from GP data (Supplementary Table 1). Similar results were seen in the supplementary analyses (Supplementary Data 9) with a slightly lower proportion of deliveries/higher proportion of losses compared to the main analyses ( ± 3%).

### Internal validation
**Deliveries.** 214,265 delivery episodes could be linked to a corresponding outcome and/or delivery-related record in at least one of the remaining three databases (excluding the database in which the episode was identified) utilised to construct the register within 2 days either side of the delivery date giving a PPV of 92%. 92% of delivery episodes derived from the HES Maternity dataset could be linked with a corresponding delivery-related procedure in the OPCS dataset. Each linked procedure occurred within 2 days of the delivery date recorded in HES Maternity. 10% (22,486) of all delivery episodes could be linked to procedures and delivery records across the GP, HES, and the OPCS datasets. Only 1% (2,565) of all delivery episodes were derived from the GP data alone, while no delivery episodes were identified from the HES APC dataset. Figure 3 (Supplementary Data 10) shows a Venn diagram of all delivery episode

data sources and whether another delivery record (or delivery-related record) was present in one or more of the other data sources within 2 days of the delivery date. The PPV for delivery-related episodes in the register increased from 58% ($n = 135,769$; Supplementary Fig. 3 and Supplementary Data 11) to 98% ($n = 229,129$; Supplementary Fig. 4 and Supplementary Data 12) from 0 to 7 days of the outcome date, respectively. 22% of deliveries could be linked to a postnatal GP record within 175 days of the delivery date. The diagrams produced for the supplementary analyses showed very similar results, with the PPV increasing from 59% to 98% from 0 to 7 days around delivery dates, respectively (Supplementary Figs. 5 –7 and Supplementary Data 13 − 15).

**Pregnancy losses.** 71% ($n = 32,799$) of pregnancy loss episodes derived from GP (51%), HES Procedures (5%), and HES APC (15%), had no corresponding loss record in any other dataset within 2 days of the loss/termination date. 23% of the remaining pregnancy loss episodes could be linked with records across two of the four data sources, with the remaining 6% of episodes linking to pregnancy loss records in three or more of the data sources (PPV = 29%). Figure 4 (Supplementary Data 16) shows a Venn diagram of all pregnancy loss episode data sources and whether another pregnancy loss record was present in one or more of the other data sources within 2 days of the loss date. The PPV for loss-related episodes in the register increased from 23% to 33% from 0 (Supplementary Fig. 8 - Supplementary Data 17) to 7 (Supplementary Fig. 9–Supplementary Data 18) days of the outcome date, respectively. The supplementary analyses produced marginally lower (but comparable) results with the PPV ranging from 20% to 31% between 0 to 7 days around pregnancy loss dates, respectively (Supplementary Fig. 10–12 and Supplementary Data 19,21).

### External validation
The register identified 138,760 livebirths among 3,034,561 eligible women during 2021, producing an age-standardised rate of 44.7 (95% CI 44.5, 45.0) livebirths per 1000 women in England. This is lower than estimates for

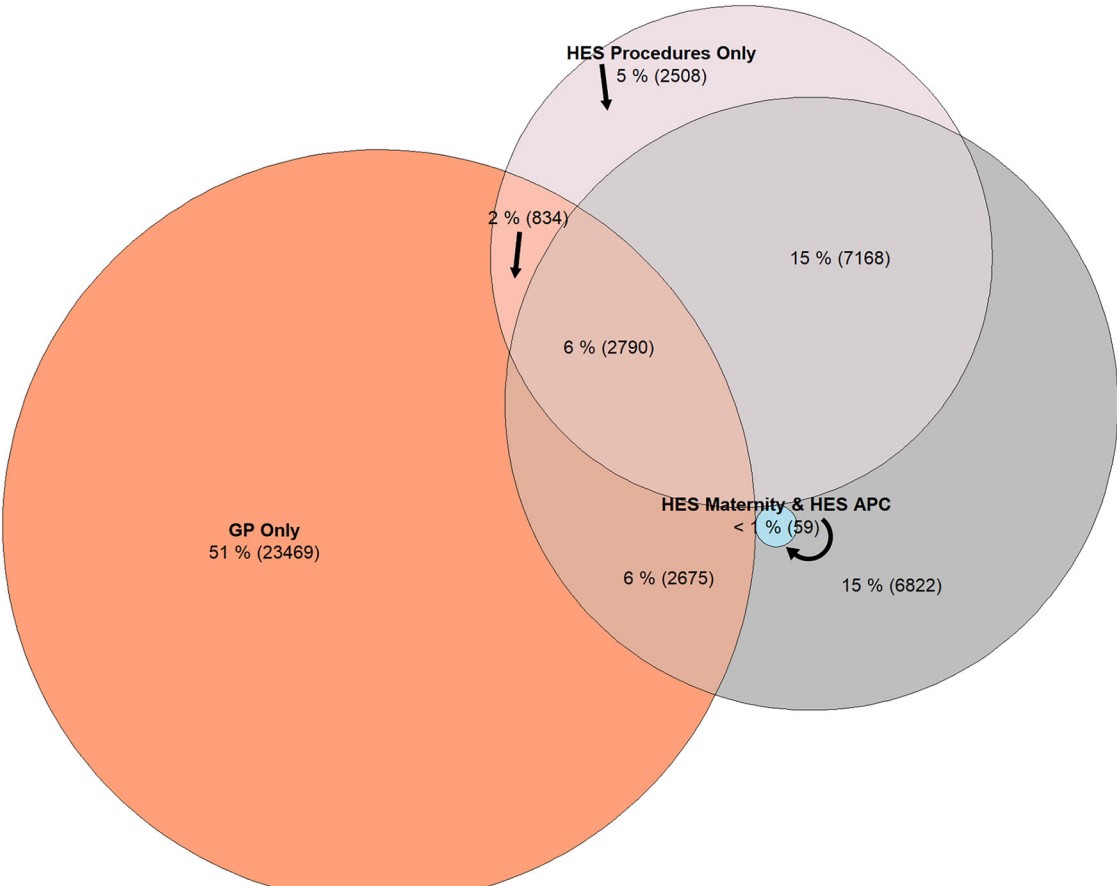

**Fig. 4 | Venn diagram—evidence of linking pregnancy loss records ($n = 46,354$)\* across data sources (±2 days of loss date)—30 December 2020 to 30 September 2022.** \*Pregnancies suppressed by the Eulerr package due to low counts or omitted because of space restrictions: GP, HES Maternity, & HES APC ($n = 15$); HES Maternity, HES Procedures, & HES APC ($n = 11$); GP, HES Maternity, HES Procedures, & HES APC ($n = 3$). Orange circle = GP; pink circle = HES Procedures; blue circle = HES Maternity; dark grey circle = HES APC. Black arrow = indicates section of diagram described by neighbouring label. Overlapping areas contain pregnancies with linking records in 1 or more datasets.

England published by the ONS (54.3 livebirths per 1000 women) for 2021[34], but is consistent with livebirth rates recorded in other registers[8]. 694 stillbirths were identified in the register, producing a marginally higher stillbirth rate of 5.0 per 1000 deliveries compared to official estimates (4.1 per 1000 deliveries)[36]. Stillbirth rates differed by ethnic group, with lower rates observed in individuals of white (4.5 per 1000 deliveries) and Chinese (0.9 per 1000 deliveries) ethnicity. Higher rates were found in those of Bangladeshi (5.1 per 1000 deliveries), Indian (5.7 per 1000 deliveries), Pakistani (6.3 per 1000 deliveries), Caribbean (7.1 per 1000 deliveries), Other Asian (7.7 per 1000 deliveries), and Black African (7.9 per 1000 deliveries) ethnicity. These figures are consistent with annual estimates published by MBRRACE (Mothers and Babies: Reducing Risk through Audits and Confidential Enquiries across the UK)[27] and ONS[34], which are listed in Table 6.

Of the pregnancy episodes identified in 2021, 9.7% ($n = 17,004$) of episodes were miscarriages, which is marginally lower than national estimates (12.5%)[37]. 8.9% of episodes were terminations ($n = 15,653$), resulting in an age-standardised termination rate of 5.0 (95% CI 4.9, 5.1) per 1000 women for 2021. This is far lower than external estimates of 18.6 per 1000 women in the population for the same year[38]. Rates of 15.5 (95% CI 15.0, 16.1), 0.35 (95% CI 0.27, 0.45), and 0.20 (95% CI 0.14, 0.27) per 1000 pregnancies in 2021 were calculated for ectopic pregnancy, molar pregnancy, and blighted ovum, respectively. This is consistent with national estimates for rates of ectopic pregnancy (11 per 1000 pregnancies)[39] and molar pregnancy (1.6 per 1000 pregnancies)[40]. No suitable published estimate could be found for the incidence of blighted ovum in the UK. All internal and external rates/proportions are available in Table 6.

The proportion in each ethnic group out of the population with delivery outcomes (livebirths and stillbirths) in our cohort compared to 2021 ONS[41] was as follows: QResearch 61% vs 70% in ONS for white women, 3.6% vs 3.6% in Indian, 3.6% vs 1.6% in Pakistani, 0.9% vs 0.8% in Caribbean and 3.7% vs 3.3% in Black African women. This shows similar proportions in most ethnic groups, but indicates a higher representation of Pakistani women in our cohort and underrepresentation of white women compared to the overall population with deliveries.

## Discussion
This paper described methods to create a unified pregnancy cohort in the QResearch database by combining individual level data from healthcare records derived from HES Maternity, GP, HES Admissions, and HES Procedures in England. The pregnancy register comprises of 266,758 women with 279,027 pregnancies recorded during the COVID-19 pandemic from 30 December 2020 to 30 September 2022, derived primarily from the HES Maternity dataset (83%) and GP records (11%). It has 232,673 deliveries during this period (comprising 99.5% livebirths and 0.5% stillbirths) and 46,354 pregnancy losses (comprising 42.4% miscarriages, 48.9% terminations and 8.7% other types of pregnancy loss). The register describes differences in pregnancy outcome by sociodemographic characteristics during the COVID-19 pandemic. Importantly, the methodology can be replicated to generate pregnancy cohorts across different follow-up periods, providing a useful resource that could be used in future research. Additionally, this work has described pregnancy outcomes by age, ethnic group, quintile of Townsend deprivation score as a measure of socioeconomic

**Table 6 | Comparison of pregnancy outcome rates in QResearch with external estimates in 2021***

| Pregnancy Outcome | PR Rate (95% CI) | PR Crude Rate | External Rate (Data Source) |
|---|---|---|---|
| Deliveries | | | |
| Live birth | 44.7 (44.5–45.0) [c d h] | 45.7 [d c] | 54.3 (ONS) [a d c] |
| Still birth | 5.0 (4.6–5.4) [b d e] | 5.0 [b d f] | 4.1 (ONS) [b d f]; 3.5 (MBRRACE) [b l f] |
| White | 4.6 (4.2–5.0) [b d e] | 4.6 [b d f] | 3.5 (ONS) [b d f]; 3.3 (MBRRACE) [b l f] |
| Indian | 5.7 (4.1–8.1) [b d e] | 5.7 [b d f] | 5.0 (ONS) [b d f]; 4.8 (MBRRACE) [b l f] |
| Pakistani | 6.3 (4.5–8.7) [b d e] | 6.3 [b d f] | 5.9 (ONS) [b d f]; 6.2 (MBRRACE) [b l f] |
| Bangladeshi | 5.1 (3.2–7.9) [b d e] | 5.0 [b d f] | 3.8 (ONS) [b d f]; 4.6 (MBRRACE) [b l f] |
| Other Asian | 7.7 (5.1–11.6) [b d e] | 7.7 [b d f] | 4.3 (ONS) [b d f]; 4.0 (MBRRACE) [b l f] |
| Black African | 7.9 (5.9–10.5) [b d e] | 7.8 [b d f] | 7.0 (ONS) [b d f]; 8.2 (MBRRACE) [b l f] |
| Caribbean | 7.1 (4.0–12.9) [b d e] | 7.1 [b d f] | 6.6 (ONS) [b d f]; 6.1 (MBRRACE) [b l f] |
| Chinese | 0.9 (0.1–6.6) [b d e] | 0.9 [b d f] | - |
| Other | 5.2 (3.9–7.0) [b d e] | 5.2 [b d f] | 4.5 (ONS) [b d f]; 5.35 (MBRRACE) [b l f] |
| Losses | | | |
| Miscarriages | 9.7 (9.6–9.8) [g] | - | 12.5 (NHS) [g] |
| Terminations | 5.0 (4.9–5.1) [d c h] | 5.1 [d c] | 18.6 (DHSC) [d c h] |
| Ectopic pregnancy | 15.5 (15.0–16.1) [e] | 15.3 [f] | 11 (Elson et al. 2016) [f] |
| Molar pregnancy | 0.35 (0.27–0.45) [e] | 0.35 [f] | 1.66 (Savage et al. 2010) [f] |
| Blighted Ovum | 0.20 (0.14–0.27) [e] | 0.20 [f] | - |

*$n$ = 175,363 pregnancies in the register from 3,034,561 eligible women in 2021.
[a]General Fertility Rate.
[b]Denominator = live births + still births.
[c]Rate per 1000 women in the population.
[d]Denominator age-range = 15–44 years.
[e]Incidence rate per 1000 pregnancy-years in the population.
[f]Rate per 1000 pregnancies in the population.
[g]Percentage of total pregnancies.
[h]Age-standardised.
[i]Ethnicity of Infant.

status[42] and region of England. In doing this, it has highlighted potential inequalities in pregnancy outcomes in this cohort.

Our internal validation of delivery and pregnancy loss episodes in the register showed a high level of agreement between data sources for deliveries, with 98% of deliveries matching with a corresponding or supporting record in another dataset that was recorded within a week of the event date. The level of agreement between data sources for pregnancy losses was lower, with only 33% of loss records linking to corresponding events recorded within 7 days of the event date. However, the proportion of deliveries (11%) and losses (17%) with linking records in both HES and primary care within 7 days were similar. The proportion of deliveries with corresponding/supporting records in HES and primary care increased to 28% when evidence of a postnatal record was also included. The low proportion of losses recorded in both individuals' GP and HES records, may be the result of individuals who have experienced a loss (excluding terminations) not engaging with the full complement of healthcare services in the event of a loss because one or another health service (GP or HES) was sought, as opposed to both.

The external validation produced results which were, generally, comparable with national and published estimates[27,43]. A greater proportion of miscarriages was captured by the register compared to terminations (9.7% vs 8.1%; 2021), but this may be expected as many terminations are carried out in specialist clinics outside of NHS hospitals. In 2021, 21% of abortions were performed in NHS Hospitals and 77–78% in independent sector clinics (which are funded by the NHS). <1% of abortions were performed privately[38]. Some terminations may also never be recorded in a patient's NHS/GP record if they opted to have the record excluded. Our results do, however, align with those found in the CPRD pregnancy register[8] in which 8.8% of pregnancies identified ended in a termination or probable termination[8]. Stillbirth rates by ethnic group were comparable to estimates published by ONS[44] and MBRRACE[43]. The proportion of stillbirths

captured by the register (0.38%) is also very similar to what was identified in CPRD Gold (0.4%) and CPRD Aurum (0.3%)[10]. It is also likely that miscarriages are underrepresented in the register because of underreporting. This may be due to miscarriages occurring very early in a pregnancy, prior to women learning they are pregnant[45]. Fear of social stigma or shame (i.e. if a woman attributes a loss to seemingly preventable lifestyle or behavioural factors) may also discourage openness/reporting[46]. Population estimates for the true rate of miscarriage range from 8% to 24% of all pregnancies[45,47,48]. The proportion of miscarriages captured by the register in the main (7%), supplementary (9.2%), and external validation (9.7%) analyses is comparable to what was identified in CPRD Gold (9.1%) and CPRD Aurum (8.4%)[10].

UK primary care records have been used to construct pregnancy records in a number of other databases[8–10,17]. However, the methods and resulting structures of existing registers differ considerably. As previously mentioned, CPRD has established pregnancy registers in two databases, CPRD GOLD and CPRD Aurum[8,10], which identify female populations who had a pregnancy at any time within the recording periods of the databases. These registers were created using GP Read codes to identify pregnancy events in the CPRD GOLD database[8] and the CPRD Aurum database[10]. Minassian et al[8]. designed an algorithm and produced a comprehensive pregnancy register routed in the CPRD GOLD database, which draws patient data from GP practices across the UK using VISION software, representing approximately 5% of the UK population[8]. The register was later extended by Campbell et al[9,10]. by creating an accompanying register in the CPRD Aurum database, which draws patient data from English practices using EMIS Web and represents 20–25% of the UK population. Both registers were validated by searching for corresponding loss and delivery records in secondary care data (HES) and comparing delivery and loss estimates to external delivery estimates reported by the ONS.

Unlike this pregnancy register, pregnancy episodes were not restricted to patients with a recorded pregnancy outcome and pregnancy episodes were not excluded if their estimated start or end dates conflicted with other documented pregnancy episodes in the CPRD register. The authors imposed minimal limitations in terms of 1) the category of pregnancy code available which was present in a patient's GP record (e.g. outcome-related Read code, delivery-related Read code) and 2) the timing of each record or outcome-related event in relation to outcomes recorded in one or more databases. This approach maximises the number of pregnancies which can be identified from the raw data. However, it increases the risk of double-counting pregnancy events through the misclassification of historic codes in a patient's GP record as new events and hence, creating pregnancy episodes with implausible start and end dates which conflict with other recorded episodes.

A register which is more comparable to the register described in this paper, was developed by Cea-Soriano et al[17,49]. using primary care records from The Health Improvement Network (THIN) database[50]. While this register was also restricted to women with a pregnancy record, it is less recent (1996–2010) and less representative of the general population than QResearch as it is considerably smaller (5% vs 20% of the UK population). A stepwise, hierarchical algorithm was used to identify pregnancy episodes based on code type. Codes which are indicative of conception (e.g. last menstrual period) were prioritised over codes which indicated a pregnancy outcome, and other pregnancy-related codes were used if neither of the other Read Code categories were available. This approach may identify many potential pregnancies as pregnancies were defined using multiple code types, creating more potential points of entry into the final cohort. This pregnancy register shows methodological similarities with the CPRD and THIN database-derived registers, while differing in a number of vital ways. The first (and most important) is that this pregnancy register utilises both hospital maternity records and GP records in its construction, with deference to NHS maternity records if there is a conflict. All other diagnostic codes (ICD-10 or SNOMED CT) derived from the remaining databases (GP, HES Admissions, HES Procedures) are afforded the same level of importance relative to each other. This allows the QResearch algorithm to easily identify and disregard non-HES Maternity records which reflect preceding pregnancy records present and better represented in patients' HES Maternity records. This dataset-based hierarchy increases the likelihood that the delivery date recorded is the true date as delivery dates recorded in hospital are typically recorded very soon after a delivery has taken place. However, in terms of primary care data, a new mother may only visit her GP practice a number of weeks after she has delivered, leading to a considerable delay between the event and the recording of the event in a patient's GP record. As an example, while 99% of deliveries identified in this cohort are derived from HES Maternity, only 10% of those deliveries show evidence of supporting outcome codes in patients' GP data recorded within a week of the hospital delivery date. In order to minimise the risk of misclassifying codes in a patient's GP records, the pregnancy register algorithm does not consider GP records to be indicative of a new pregnancy if they conflict with pregnancies recorded in the HES Maternity database or a preceding pregnancy which has already been identified. This more conservative approach restricts the number of potential pregnancies but minimises the risk of misclassifying records derived from patients' GP records.

This study has several strengths. The dataset was based on the QResearch GP database linked to HES datasets which allowed for the verification of outcomes using multiple data sources. It has developed an algorithm to identify pregnancies, determine their outcome, and set up linkages which will facilitate creating pregnancy cohorts with different follow-up periods for future research studies. This will allow future investigations into the associations between potentially adverse exposures and pregnancy outcomes, including rare postnatal outcomes such as congenital anomalies. The cohort defined in the development of the register, will be used to investigate post-marketing vaccine safety and effectiveness in pregnant women for the COVID-19 vaccines, but could, theoretically, be used for research into any

vaccination programme. A further strength of this study was the approach used to identify consistencies in records between the four data sources. This approach improved the validity and reliability of the algorithm used in creating the pregnancy cohort. For instance, delivery episodes matched with outcome records in at least one additional database (the database they were derived from plus 1 other) for 98% of deliveries within a week of the delivery date. This high level of agreement between data sources supports the accuracy and sensitivity of the algorithm. Grouping the cohort into pregnancy outcomes, delivery outcomes, and types of pregnancy losses by demographic and socio-economic characteristics has enabled the assessment of the cohort's representativeness through comparison with the general population. We have reported pregnancy outcomes from different ethnic and social groups and from different regions of England, demonstrating good national and social coverage. This has revealed some potential discrepancies in pregnancy outcomes amongst women from different population groups, which warrant further investigation in future research.

The study and structure of the algorithm used to establish the register, while robust, has several limitations. Almost all pregnancy losses and a small number of deliveries received an approximated estimated date of conception and length of gestation. These approximations may slightly under or overestimate the length of each pregnancy. Pregnancy loss codes which could not be linked to a specific outcome (e.g. ICD-10 code O06—Unspecified abortion, which may represent a miscarriage or termination) were categorised as terminations in accordance with the register's prioritisation process. HES Maternity records without an outcome were excluded prior to creation of the register. However, the number of records affected was small. Of the 231,871 records available, less than 1% (1,321) were dropped for this reason. Length of gestation values derived from the relevant HES Maternity fields (GESTAT_n) may also be slightly over/underestimated, as these values were stored as and converted from whole weeks to days. It is likely that many of these values were rounded up/down prior to inclusion in HES Maternity.

Pregnancies which may have been ongoing during the study period, but which did not have a pregnancy outcome, where not included in the register. The register's algorithm allows for the inclusion of pregnancies which commenced prior to the study start date, but due to the importance afforded to the presence of an outcome code, pregnancies which may have occurred but did not end during the study period where not eligible or identifiable. If vaccine safety is the primary exposure of interest, then careful selection of the dates of the period of exposure can be undertaken to ensure that any related outcomes are captured. Lastly, it is evident from the supplementary analyses that the lack of primary care data for the last 8 months of the study period may have caused outcomes which are more likely to be GP data dependent (e.g. miscarriages) to be slightly underrepresented during the study period. However, as HES OPCS is also a rich source of pregnancy loss data, the impact on the proportion of losses in the full cohort (16.6%) compared to the restricted (19.6%) appears to be small. The level of agreement between GP and HES ethnicity data was not assessed in this study as it was beyond its scope. However, previous research analysing the level of agreement been ethnicity recording in HES and primary care data in CPRD, has reported a high level of agreement (93.3%) between the two databases. Agreement was lowest in those of "mixed" and "other" ethnicity[51]. While the QResearch database can be considered to be representative of the English population, not all regions are equally well represented. QResearch data is slightly less representative of the populations in the East Midlands, East of England, and North East, due to fewer participating GP practices in those regions.

The pregnancy register represents a cohort of women who were pregnant during the emergency phase of the COVID-19 pandemic (December 2020 to September 2022), but the methods can be easily replicated to define a similar cohort over a different period, as has been demonstrated in the supplementary analyses. In creating the register, we have provided an opportunity for researchers who want to use the QResearch database to request data on cohorts of pregnant women for future analyses. This resource will make it easier to establish patterns of healthcare use and determine

associations between health conditions or post-approval medication use, and health outcomes for mothers and babies in a large population. Importantly, it will make it possible to investigate ethnic, regional, and social (deprivation-related) differences, both in medication use and in pregnancy outcomes. This will provide evidence that can inform individual guidance for pregnant women as well as their infants. However, investigators planning on using the register to conduct studies assessing drug safety or medication use during pregnancy should be mindful that the database's prescription data, while comprehensive, is not a measure of adherence. Common methods such as the Medication Possession Ratio (MPR)[52] and Proportion of Days Covered (PDC)[53] are also not viable, as both require dispensing data which QResearch does not contain. While evidence on adherence is scarce, under-ascertainment or misclassification of medication among patients is a possibility, with some evidence suggesting that between one third to half of all medication prescribed for long term conditions is taken improperly[54]. This may increase the likelihood of under/overestimating the safety and/or effectiveness of medications used during pregnancy.

Now that a register has been established, the research team will proceed with examining the effectiveness and safety of COVID-19 vaccination during pregnancy in this cohort. Future linkages are planned with other datasets including the Maternity Services Dataset (MSDS) 1 & 2[25] and HES Outpatients[22]. These datasets contain additional clinical codes, including an EDD code for estimated date of delivery based on ultrasound scan data in the MSDS dataset, which will improve certainty in the determination of outcomes, as well as pregnancy start and end dates. MSDS also contains pregnancy data (booking appointment, etc.) regardless of whether a pregnancy has concluded or not. This will allow for the inclusion of pregnancies which may be ongoing at the end of any study's observation period. Additionally, it will provide a deterministic mother-baby link, which will allow for the study of associations between exposures in pregnancy and outcomes in children.

## Data availability

All eligible persons/researchers may gain access to the QResearch database upon successful completion of a project application and receipt of ethical approval from the QResearch Scientific Committee. However, due to national and organisational data privacy regulations, the data (ONS, HES, and GP) that supported the construction of the algorithm are not publicly available as they are based on pseudonymised national clinical records. Further details on database access requirements, eligibility, costs, etc., may be found on the QResearch Website: https://www.qresearch.org. The QResearch database is housed at Queen Mary University of London. Supplementary Data 10 contains the source data for Fig. 3 and Supplementary Data 16 contains the source data for Fig. 4. Descriptions of Supplementary Data 1-9, 11-15, and 17-20 may be found in the supplementary file entitled, Description of Supplementary files.

## Code availability

Sample STATA code is available online from the following repository: https://doi.org/10.5281/zenodo.16635465[55] This code may be used to inform the creation of similarly structured registers.

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

## Acknowledgements

We acknowledge the contribution of general practices who contribute to QResearch and Queen Mary University of London for expertise in developing or supporting the QResearch database. This project involves data derived from patient-level information collected by the NHS, as part of the care and support of cancer patients. The hospital, cancer and mortality data are collated, maintained and quality assured by the National Disease Registration Service which is part of NHS England. Access to the data was facilitated by the NHS England Data Access Request service. NHS England bears no responsibility for the analysis or interpretation of the data. This study is funded by the National Institute for Health and Care Research (NIHR) School for Primary Care Research (project reference: 591). The views expressed are those of the authors and not necessarily those of the NIHR or the Department of Health and Social Care.

## Authors contributions

J.H.C. obtained funding, obtained data approvals, contributed to interpretation of the analysis, reviewed the first and final drafts of the paper. A.J.H.L.S. led onthe writing of the manuscript, the design of the pregnancy algorithm, the data management, and the internal/external validation. He

also contributed to the data analyses, study design, and the interpretation of the results. C.C. contributed to the analysis, interpretation of the results and reviewed the manuscript drafts. J.A.H. contributed to funding acquisition, study design, data management, data analysis and interpretation. E.C. contributed to the data management, data analysis and reviewed the draft manuscript. T.R. contributed towards the data management and the writing of the of the manuscript. W.M.M. contributed towards the data management, data analysis, and the writing/revision of the manuscript. W.M. contributed towards the data management, data analysis, design of the algorithm, and writing of the manuscript. A.S. contributed towards the study design, design of the algorithm, and interpretation of the results. Q.P.C. contributed towards the study design/conception, design of the algorithm, and review of the final manuscript. All authors contributed to the critical revision of the manuscript and approved the final version of the manuscript.

## Competing interests

The authors declare the following competing interests: A.S. is a member of the Scottish Government Chief Medical Officer's COVID-19 Advisory Group, the Scottish Government's Standing Committee on Pandemics and Astra-Zeneca's Thrombotic Thrombocytopenic Advisory Group. All roles are unremunerated. J.H.C. reports grants from National Institute for Health Research (NIHR) Biomedical Research Centre, Oxford, grants from John Fell Oxford University Press Research Fund, grants from Cancer Research UK (CR-UK) grant number C5255/A18085, through the Cancer Research UK Oxford Centre, grants from the Oxford Wellcome Institutional Strategic Support Fund (204826/Z/16/Z) and other research councils, during the conduct of the study. J.H.C. is an unpaid director of QResearch, a not-for-profit organisation which is hosted by Queen Mary University of London. J.H.C. is a founder and shareholder of ClinRisk Ltd and was its medical director until 31st May 2019 and shareholder until Aug 2023 when 100% of the company was donated to Endeavour Health Care Charitable Trust. ClinRisk Ltd produces open and closed source software to implement clinical risk algorithms (outside this work) into clinical computer systems. J.H.C. is chair of the NERVTAG risk stratification subgroup and a member of SAGE COVID-19 groups. All remaining authors declare no competing interests related to this paper.

## Additional information

## QResearch Pregnancy Consortium

Marian Knight[4], Kenneth Hodson[5], Anthony Harnden[2], Jonathan Van-Tam[3], Carol Dezateux[1], Brenda Kelly[6], Alessandra Morelli[7], Joanne Enstone[3], Sharon Dixon[2] & Aziz Sheikh[2]

[4]Nuffield Department of Population Health, University of Oxford, Oxford, UK. [5]Faculty of Medical Sciences, Newcastle University, Newcastle, UK. [6]Oxford University Hospitals NHS Foundations Trust, Oxford, UK. [7]Nuffield Department of Clinical Neurosciences, University of Oxford, Oxford, UK.

