## [Transparent Peer Review file · Communications Medicine]

Methods to establish a Pregnancy Register in the QResearch Database

Corresponding Author: Dr Jennifer Hirst

Version 0:

Decision Letter:

**** Please ensure you delete the link to your author homepage in this email if you wish to forward it to your coauthors ****

Dear Dr Hirst,

Your manuscript entitled "Establishing the QResearch Pregnancy Register in the QResearch® database: methods and cohort description" has now been seen again by our referees, whose comments appear below. In light of their advice I am delighted to say that we are happy, in principle, to publish a suitably revised version in Communications Medicine.

We therefore invite you to revise your paper one last time to address the remaining concerns of our reviewers. At the same time we ask that you edit your manuscript to comply with our format requirements and to maximise the accessibility and therefore the impact of your work.

* Please see the attached document for editorial requests for the final version (.docx file). Please confirm that the necessary changes have been made, and clearly indicate where each change appears in the manuscript by using the right-hand column in the document. The completed document must be uploaded as a related manuscript file.

* Please review our [final submission file checklist](https://www.nature.com/documents/commsj-file-checklist.pdf) to ensure all necessary files are present with your final submission and to avoid delays in accepting your manuscript.

It is important that you pay careful attention to the requests in these documents to avoid a delay in formal acceptance of the article.

Open access

"Communications Medicine is a fully open access journal. Articles are made freely accessible on publication. For further information about article processing charges, open access funding, and advice and support from Nature Research, please visit <https://www.nature.com/commsmed/open-access>

Please use the following link to upload your revised files:

Link Redacted

We hope to hear from you within two weeks; please let us know if the process may take longer.

Congratulations on an excellent paper!

Best regards,

Meg Mashbat, PhD
Senior Editor
Communications Medicine

PS: At acceptance, the corresponding author will be required to complete a Licence to Publish on behalf of all authors, declare that all required third party permissions have been obtained and provide billing information in order to pay the article-processing charge (APC) via credit card or invoice. Please note that your paper cannot be sent for typesetting to our production team until we have received these pieces of information; **therefore, please ensure that you have this information ready when submitting the final version of your manuscript.**

REVIEWERS' COMMENTS:

Reviewer #1 (Remarks to the Author):

The comments from the review of the manuscript have been addressed adequately. No additional comments.

Reviewer #4 (Remarks to the Author):

Thank you to the authors of this important paper for their thoughtful revisions. I enjoyed reading it again, as well as the discussion in the response. I only have a few further, very minor comments.

The generation of pregnancy episodes in this register is clear and would allow researchers subsequently using a cohort from the register to understand (as clearly as is possible with real-world pregnancy data) the origin of each episode. They have taken inspiration from other pioneers in this field, Minassian, Campbell, and others, to build another important resource for pregnancy research. The comparability with other registers and national statistics provides a level of reassurance that this is a representative sample.

Looking forward to seeing the updates with MSDS and future iterations and improvements!

Typographical errors

Page 2 line 35 "will further facilitate research in pregnancy"

Page 2 line 48 define APC

Page 2 line 49 define OPCS

Page 3 line 89 "Nordic registries" (not registers)

Page 3 line 100 "QRResearch database, named the QRResearch Pregnancy Register."

Page 4 line 104 "uptake, safety and effectiveness"

Page 4 line 126 define ICD

Page 5 line 175 "of less than 154"

Page 9 313 "in the available data sources" 314 "deduce which data source"

Consistency with live birth or livebirth

Page 10 line 384 "0.5%"

Bottom of 10 and top of 11 needs proofing

Queries

How did you ascertain the length of gestation in days from HES (page 7 line 242)? Gestat in HES Maternity is given in weeks (as stated in the discussion)

Page 13 line 496 the reported percentage of TOP / probable TOP in the reference you've cited isn't right - it's 8.8% not 6.9% in the CPRD Pregnancy Register

Page 13 line 497-498 may be of interest to say that the stillbirth estimate is not in line with CPRD but in line with other estimates to bolster it's utility alongside other registers

Statement at the end of the methods stating which programming tool was used for data analysis would be nice (I know R is mentioned above but referring to a specific viz)

I know that unknown outcomes from unfinished pregnancies are mentioned in the limitations, but perhaps an additional statement about the limitation of other unknown outcome pregnancies (as seen in the CPRD Pregnancy Register). Of course, the approach taken here to only include pregnancies that have an outcome is valid (and implemented in other EHR pregnancy registers), but there will be pregnancies that are reported to the GP and are subsequently lost early in gestation (for example, and make up some of the unknown outcome pregnancies in CPRD); these may not be reported to the doctor and are therefore missed. Inevitable but a broad limitation of this type of work that might be worth a mention in the discussion.

Table 2 - the <5 in ethnic group 3+ pregnancies can still be calculated? I can't see any obscured counts to protect against this

Curious as to why operative deliveries aren't captured in HES procedures - is it a data capture artefact?

Florence Z. Martin (University of Bristol and IQVIA)

** See Nature Portfolio's author and referees' website at www.nature.com/authors for information about policies, services and author benefits

Version 1:

Decision Letter:

** Please ensure you delete the link to your author homepage in this email if you wish to forward it to your coauthors **

Dear Dr Hirst,

Thank you once again for submitting your manuscript, "Methods to establish a Pregnancy Register in the QResearch® Database," to Communications Medicine. It has now been seen again by the in-house editorial team. The concerns of our reviewers have now been addressed, but there are some amendments needed before we can accept your paper.

We ask that you edit your manuscript according to list of requests from the team.

From the Editor:

1. Thanks for providing a draft Plain Language Summary. It reads well in most parts, but please simply, briefly explain or remove the following term to improve clarity of reading for the general public: QResearch database
2. Please reformat the labels for the Venn diagrams in Figures 3 and 4, to indicate which label refers to which section of the diagram. If the label does not fit within the designated section, please use lines as a pointer to indicate.
3. Thank you for providing a detailed Data availability statement. However, please remove all information related to the description of supplementary data files from this section and submit it as a related manuscript file titled "Description of Supplementary files". For the source data, please leave the statement that describes the source data for Figures 3 and 4.
4. Please move Code Availability section directly after Data Availability.

To allow us to move forward with your work, we also ask that you edit your manuscript according to the attached table. Please read this document carefully as we will be unable to further assess your revised paper until these important points are addressed.

Please outline all revisions made in the right-hand column and return the completed table with your updated manuscript files as a Related Manuscript file.

Please use the link below to submit your revised files:

Link Redacted

When resubmitting, please provide a clean version of your paper and related files once again.

We hope to receive this updated version of your paper within 1 week, but please let us know if you find that you need more time.

Best regards,
Meg Mashbat, PhD
Senior Editor
Communications Medicine

Version 2:

Decision Letter:

**** Please ensure you delete the link to your author homepage in this email if you wish to forward it to your coauthors ****

Dear Dr Hirst,

Thank you once again for submitting your manuscript, "Methods to establish a Pregnancy Register in the QResearch® Database," to Communications Medicine. The concerns of our reviewers have now been addressed, but there is one amendment needed before we can accept your paper.

1. For all research involving human participants, informed consent to participate in the study should be obtained from participants (or their parent or legal guardian in the case of children under the age of 16) and a statement to this effect should appear in the manuscript. If the need for informed consent was waived by the ethics committee, please state this and the reason clearly.

Please provide responses to above requests and upload the file as 'Related Manuscript File' upon re-submission.

Please use the link below to submit your revised files:
<https://mts-commsmed.nature.com/cgi-bin/main.plex>

When resubmitting, please provide a marked-up manuscript with all changes highlighted, as well as a clean version of your paper.

We hope to receive this updated version of your paper within 1 week, but please let us know if you find that you need more time.

Best regards,
Meg Mashbat, PhD
Senior Editor
Communications Medicine

Reviewers' comments:

Version 3:

Decision Letter:

Dear Dr Hirst,

We are delighted to accept your manuscript titled "Methods to establish a Pregnancy Register in the QResearch Database" for publication in Communications Medicine. Thank you for choosing to publish your interesting work with us.

The Editor made slight changes to your final article file before acceptance. This includes numbering your equation and changing your alphabetically numbered list on page 7 to bulleted list, to ensure it fits our formatting requirements.

Licence to Publish and Article-Processing Charge

In approximately 7-10 business days you will receive an email with a link to choose the grant of rights necessary for publishing your paper and – if applicable – to provide payment information for your article-processing charge (APC), either via credit card or by requesting an invoice.

If needed, our Author Services team will be in touch regarding any additional information that may be required.

In order to avoid any delays, please ensure that you have emails from Springer Nature whitelisted in your mail system.

If you have any questions about our publishing options, costs, Open Access requirements, or our legal forms, please contact

ASJournals@springernature.com

We will edit your manuscript to ensure that it conforms with our house style and send you a link to an online eProof for checking in a separate email to the publishing agreements. Please read your proof with great care to ensure that the meaning has not been altered. We also suggest you discuss the proof with your co-authors, but please ensure that only one author communicates with us and that only one set of corrections is returned via the online correction in the eProof. The corresponding (or nominated) author is responsible on behalf of all co-authors for the accuracy of all content, including spelling of names and current affiliations.

To ensure prompt publication, we request that proofs are returned within two working days. If there is any period within the next four weeks in which you won't be available, please nominate a co-author with whom we can correspond, and let us know their email address as soon as possible.

Please note that production will not continue until the Licence to Publish and Article-Processing Charge steps are completed and your proof corrections are submitted.

Please note that your Supplementary Information files may have been edited for style and are now finalized. They will be uploaded directly to the Communications Medicine website in preparation for publication of the Article. Any requests to make changes will only be considered in exceptional circumstances and will result in a delay to publication.

Acceptance of your manuscript is conditional on all authors' agreement with our publication policies (see www.nature.com/nature/authors/policy/index.html). In particular, your manuscript must not be published elsewhere and there must be no announcement of the work in the media until the publication date. At this stage, you may wish to make your institution's press office aware of the forthcoming publication, if you wish to bring your work to the media's attention, so that they can start preparing any publicity. Please note that the paper is still under embargo until it is published in the journal. Further details of our embargo policy can be found here <http://www.nature.com/authors/policies/embargo.html>.

Publication is typically within two to three weeks of acceptance. Please note there will be no further correspondence about your publication date. When your article is published, you will receive a notification email. **If you are planning an embargoed press release or require a specific publication date, please complete our [scheduling requests form](https://forms.office.com/e/ed7NBDDd08u), or contact commsproduction@springernature.com, as soon as possible after acceptance and we will endeavour to accommodate your request.** For further information on the journey of your article from acceptance to publication, please see our [Author FAQs](https://www.nature.com/documents/Author_FAQs.pdf).

We may also promote the publication of your paper via press release or through our social media channels or other promotional campaigns. If you would like us to include any Twitter handles in our Twitter posts (such as those of the authors or their institutions), please contact me. If you post any media, including social media posts, blog posts, or videos, related to your manuscript that you would like us to share, we would be happy to do so.

If you have not already done so, we would welcome the submission of material for the 'Featured Image' section on the Communications Medicine home page. Images should relate to the content of your manuscript, but need not be contained within the paper. Suggestions should be sent by email to commsmed@nature.com. Please provide 1400x400-pixel RGB images. Unfortunately, we cannot promise that your suggestions will be used.

Providing great service is very important to us. We would greatly appreciate any comments you have about your experience at Communications Medicine. I hope that we have been able to meet your expectations and look forward to working with you again in the future.

Thank you again for the opportunity to publish your work in Communications Medicine!

Best regards,

Meg Mashbat, PhD
Senior Editor
Communications Medicine

** See Nature Research's author and referees' website at www.nature.com/authors for information about policies, services and author benefits

A form to order reprints of your Article is available at <http://www.nature.com/reprints/author-reprints.html#options>

If you have any questions about open-access invoicing or payment, please contact authororders@nature.com

You can now use a single sign-on for all your accounts, view the status of all your manuscript submissions and reviews,

access usage statistics for your published articles and download a record of your refereeing activity for the Nature Research journals.

REVIEWERS' COMMENTS:

Reviewer #1 (Remarks to the Author):

The comments from the review of the manuscript have been addressed adequately. No additional comments.

Reviewer #4 (Remarks to the Author):

Thank you to the authors of this important paper for their thoughtful revisions. I enjoyed reading it again, as well as the discussion in the response. I only have a few further, very minor comments.

The generation of pregnancy episodes in this register is clear and would allow researchers subsequently using a cohort from the register to understand (as clearly as is possible with real-world pregnancy data) the origin of each episode. They have taken inspiration from other pioneers in this field, Minassian, Campbell, and others, to build another important resource for pregnancy research. The comparability with other registers and national statistics provides a level of reassurance that this is a representative sample.

Looking forward to seeing the updates with MSDS and future iterations and improvements!

Typographical errors

Page 2 line 35 "will further facilitate research in pregnancy"

Thank you. The text has been updated.

Page 2 line 48 define APC

Thank you. APC has been removed from the abstract and replaced with "Admissions" for the sake of word count. The acronym, APC, is described in the "Data sources" section of the paper, as well as the use of "HES Admissions" as a form of shorthand for the dataset.

Page 2 line 49 define OPCS

Thank you. OPCS has been removed from the abstract and replaced with "Procedures" for the sake of word count. The acronym, OPCS, is described in the "Data sources" section of the paper, as well as the use of "HES Procedures" as form of short hand for the dataset.

Page 3 line 89 "Nordic registries" (not registers).

Thank you. The text has been amended.

Page 3 line 100 "QResearch database, named the QResearch Pregnancy Register."

Thank you. The relevant text has been removed as authors have decided to refer to the aforementioned "QResearch Pregnancy Register" simply as the "the register" or "this register".

Page 4 line 104 "uptake, safety and effectiveness"

Thank you. The text has been amended.

Page 4 line 126 define ICD

Thank you. Please see updated text below:

"Electronic clinical code lists from primary care records (SNOMED CT), hospital diagnoses (International Statistical Classification of Diseases and Related Health Problems, 10th Revision (ICD-10))"

Page 5 line 175 "of less than 154"

Thank you. The text has been amended.

Page 9 313 "in the available data sources" 314 "deduce which data source"

Thank you. The text has been amended.

Consistency with live birth or livebirth

Thank you. All relevant text amended to "livebirth".

Page 10 line 384 "0.5%"

Thank you. The text has been amended.

Bottom of 10 and top of 11 needs proofing

Thank you. The bottom of 10 and top of 11 have been proofread and edited in a number of places. See text below:

"231,619 deliveries resulted in a livebirth (99.5%) and 1,054 in stillbirths (0.5%). Characteristics of women with a delivery outcome are shown in Table 4. Proportions of stillbirths were similar across all age groups and quintiles of deprivation, with the highest proportion among those in the oldest (40 to 50; 0.6%) and most deprived (Quintile 5; 0.6%) groups. The largest proportions of stillbirths were amongst those of

Caribbean (0.7%), Black African (0.7%), and Other Asian (0.7%) ethnicity, with the lowest in those of Chinese (0.3%) and White (0.4%) ethnicity. Yorkshire & Humber (0.6%) was the region with the highest proportion of stillbirths compared to the East of England (0.3%), which had the lowest. The supplementary analyses (Supplementary Data 7) produced very similar results, with identical proportions of deliveries resulting in a livebirth (99.5%) and stillbirth (0.5%) compared to the main analyses.

Characteristics of the populations with different pregnancy loss outcomes are shown in Table 5. Pregnancy losses consisted of 19,648 (42.4%) miscarriages, 22,655 (48.9%) terminations, and 4,051 (8.7%) other pregnancy losses (ectopic pregnancies, molar pregnancies, and blighted ovum cases). The highest proportion of terminations (67.3%) and the lowest proportion of miscarriages (28.5%) were found in the 15-20 age group. The highest proportion of miscarriages was observed in those in the 40-50 age group. In terms of ethnicity, the highest proportions of miscarriages were amongst those of Pakistani (57.3%) and Bangladeshi (54.2%) ethnicity, and lowest in those of Caribbean (37.1%) ethnicity. The proportion of miscarriages was higher than the proportion of terminations in those with missing deprivation data (46.4% vs 44.8%) and in the East of England (47.7% vs 42.9%). All other deprivation groups and regions of England had higher proportions of terminations compared to miscarriages. The highest proportions of terminations were amongst those of White (50.4%), Chinese (51.1%), Caribbean (49.2%), and Black African (48.1%) ethnicity. High proportions were also present for those of Other (47.9%) and no recorded ethnicity (55.8%), as well as those in the North-East (62.1%) region compared to other parts of England. The proportion of pregnancy losses which were neither a miscarriage nor termination (other pregnancy losses), was similar by ethnic group (8.4% to 10.3%) apart from those of Caribbean (14.9%) ethnicity. The supplementary analyses (Supplementary Data 8) produced comparable results. However, the highest proportion of pregnancy losses were miscarriages (47%) with a marginally lower proportion of terminations (45%). Other pregnancy losses were similar to the main analyses with a proportion of 8%. The proportion of miscarriages was similar by ethnic group with the highest proportion remaining in those of Pakistani ethnicity (62.1%). The distribution of miscarriages and terminations were similar across all other covariates (age group, level of deprivation, and region) with a 3-5% increase in miscarriages and a commensurate drop in the proportion of terminations.”

Queries

How did you ascertain the length of gestation in days from HES (page 7 line 242)? Gestat in HES Maternity is given in weeks (as stated in the discussion).

Thank you for the question. The length of gestation in days is calculated by multiplying the number of weeks of gestation by 7. Text in the methods has been amended, see below:

“Length of Gestation (HES field or GP code): estimated date of conception = delivery date minus Length of gestation in days. (Preference was given to HES if both were present). The HES field (GESTAT_n) was converted from whole weeks to days (weeks of gestation x 7).”

Page 13 line 496 the reported percentage of TOP / probable TOP in the reference you've cited isn't right - it's 8.8% not 6.9% in the CPRD Pregnancy Register

Thank you very much! The percentage has been corrected.

Page 13 line 497-498 may be of interest to say that the stillbirth estimate is not in line with CPRD but in line with other estimates to bolster it's utility alongside other registers

Thank you for the suggestion. The following text has been added commenting on the total proportion of stillbirths present in the register compared to CPRD Gold and CPRD Aurum, as a stillbirth rate per 1000 live and stillbirths is not reported in the paper. See text below:

“The proportion of stillbirths captured by the register (0.38%) is also very similar to what was identified in CPRD Gold (0.4%) and CPRD Aurum (0.3%).[8]”

Statement at the end of the methods stating which programming tool was used for data analysis would be nice (I know R is mentioned above but referring to a specific viz)

Thank you. The following text has been added to the methods section:

“All the main tables were produced and the data management/analyses conducted using Stata MP, version 18.”

I know that unknown outcomes from unfinished pregnancies are mentioned in the limitations, but perhaps an additional statement about the limitation of other unknown outcome pregnancies (as seen in the CPRD Pregnancy Register). Of course, the approach taken here to only include pregnancies that have an outcome is valid (and implemented in other EHR pregnancy registers), but there will be pregnancies that are reported to the GP and are subsequently lost early in gestation (for example, and make up some of the unknown outcome pregnancies in CPRD); these may not be reported to the doctor and are therefore missed. Inevitable but a broad limitation of this type of work that might be worth a mention in the discussion.

Thank you. The following text has been added to the discussion:

“It is also likely that miscarriages are underrepresented in the register because of underreporting. This may be due to miscarriages occurring very early in a pregnancy, prior to women learning they are pregnant.[42] Fear of social stigma or shame (i.e. if a woman attributes a loss to seemingly preventable lifestyle or behavioural factors) may also discourage openness/reporting.[43] Population estimates for the true rate of miscarriage range from 8% to 24% of all pregnancies.[42, 44, 45] The proportion of miscarriages captured by the register in the main (7%), supplementary (9.2%), and external validation (9.7%) analyses is comparable to what was identified in CPRD Gold (9.1%) and CPRD Aurum (8.4%).[8]”

Table 2 - the <5 in ethnic group 3+ pregnancies can still be calculated? I can't see any obscured counts to protect against this.

Thank you. It should not be possible to calculate the precise value for the censored count as a range is provided instead of a count for the “2 pregnancies” column on its left. Please see relevant row below:

“| Chinese | 2,149 (97.0%) | [62-65] * | <5** | 2,215 |”

Curious as to why operative deliveries aren't captured in HES procedures - is it a data capture artefact?

Thank you for the question. Operative deliveries are indeed captured in the HES Procedures data; however, they do not provide the required level of detail regarding the outcome of the pregnancy (livebirth, etc.) which was needed for defining the pregnancy episodes. However, these non-specific delivery codes were used as part of the internal validation.

Florence Z. Martin (University of Bristol and IQVIA)